# Enhancing Treatment Effect Estimation with Generation-Driven Data Augmentation

## Abstract

We introduce GATE, a framework for improving the estimation of conditional average treatment effects (CATE) from observational data. Our framework leverages generative models to selectively augment datasets with synthetic potential outcomes, thus addressing the covariate shift problem inherent in CATE estimation. Crucially, GATE enables the integration of external knowledge into downstream CATE models, by leveraging generative models trained on external data sources, such as large language models (LLMs). These models utilize rich contextual information, such as dataset metadata, to generate synthetic potential outcomes grounded in real-world contexts. While generative models can introduce bias when imperfect, we theoretically demonstrate that restricting augmentation to a carefully chosen subsets of the covariate space can allow to achieve performance gains despite these imperfections. Empirically, GATE instantiated with LLMs consistently improves a wide range of CATE estimators, narrowing performance gaps between learners and underscoring the advantages of incorporating external knowledge through generative augmentation, particularly in small-sample regimes.

## 1 Introduction

**Motivation.** Conditional Average Treatment Effect (CATE) estimation has received a lot of attention from the machine learning (ML) literature (Shalit et al., 2017; Künzel et al., 2019; Kennedy, 2023). This should not be a surprise: tools allowing to leverage *observational data* to estimate the person-alised effects of interventions are an invaluable asset in domains varying from healthcare (Gershon et al., 2021) to economics (Baum-Snow & Ferreira, 2015) or marketing (Hill et al., 2015). However, despite its immense benefits, estimating CATE from observational data remains challenging.

In observational data, especially in high-stakes domains, treatments are assigned non-randomly. This non-random treatment assignment, particularly when guided by expert judgment (Hüyük et al., 2024), leads to a *covariate shift*, where covariate distributions differ between treated and control groups, leading to unreliable or high-variance treatment effect estimates (Johansson et al., 2022). Covariate shift can have especially negative consequences in *small-sample regimes* (Alaa & van der Schaar, 2018), prevalent in case of novel treatments, rare diseases or data collected at the country level.

Existing approaches address the covariate shift through *model* specification, employing inverse-propensity weighting (Abrevaya et al., 2015), representation learning (Johansson et al., 2016; Shalit et al., 2017), or both (Assaad et al., 2021; Hassanpour & Greiner, 2019). In this work, we explore a *complementary solution* to the covariate shift problem, which can be used alongside *any* CATE learner: addressing covariate shift by manipulating the *dataset* rather than the model.

**Generative Augmentation for CATE.** To this end, we introduce GATE (*Generative Augmentation for Treatment Effect estimation*), a flexible data augmentation framework that leverages diverse generative models to supercharge the performance of existing CATE estimators. By augmenting[1] the observational dataset with missing potential outcomes sampled from a generative model, GATE not only *increases the effective sample size* (as is typical in data augmentation) but also directly *mitigates the covariate shift* inherent in CATE estimation. Importantly, as we demonstrate both theoretically and empirically, the reduction in covariate shift can counterbalance biases introduced by a potentially

---

[1] We refer to this solution as augmentation, rather than imputation, as we approach the problem from the perspective of the dataset $\mathcal{D}_t^{(\mathrm{obs})}$, rather than $\mathcal{D}^{(\mathrm{obs})}$ (see Section 2 for definitions).

imperfect generative model. This distinguishes GATE from traditional data augmentation, where inaccuracies in the generative model can easily degrade performance (Manousakas & Aydöre, 2023).

To maximize effectiveness, GATE restricts augmentation to a carefully selected subset of the covariate space, the *admissible set*, where the generative model's predictions are expected to be most reliable. This principled selection process, informed by the properties of the generative model, ensures robust augmentation and minimizes the risk of introducing noise.

**Leveraging External Knowledge Through Generative Models.** Beyond mitigating covariate shift, data augmentation offers a unique opportunity to inject external knowledge into downstream CATE models. GATE achieves this by leveraging generative models trained on external data sources, contrasting imputation methods limited to local regression modules or GANs (Aloui et al., 2023; Yoon et al., 2018). In particular, we propose to instantiate GATE with *Large Language Models* (LLMs), which benefit from extensive pretraining and have been shown to improve performance of downstream models across diverse tasks (Choi et al., 2022; Zhu et al., 2023; Seedat et al., 2024).

The key advantage of using LLMs as generative models lies in their ability to leverage rich metadata present in observational datasets, such as textual descriptions of covariates or contextual information. By utilizing this metadata, LLMs can generate potential outcomes that align with common-sense reasoning, grounding downstream CATE models in real-world contexts. Recent studies also highlight LLMs' ability to uncover causal structures in real-world data (Richens & Everitt, 2024; Zečević et al., 2023; Long et al., 2024), further validating their use in this context.

However, there are valid concerns regarding the safety and robustness of employing LLMs in the causal setting, particularly given their propensity for hallucinations. As such, we approach the use of LLMs with a critical and cautious stance, acknowledging the uncertainties and potential pitfalls. Through empirical experiments, we investigate under which circumstances LLMs outperform other generative models within our framework. Our results indicate that LLMs can be particularly beneficial in low-data regimes, and under a strong covariate shift.

> **Contributions.** ① **Data augmentation framework for CATE:** We propose GATE, a flexible data augmentation framework that leverages diverse generative models to obtain missing potential outcomes, addressing the challenges of covariate shift in CATE estimation. Unlike existing approaches for potential outcome imputation, GATE can seamlessly incorporate insights from external data sources to enhance downstream CATE estimators. ② **Theoretical analysis:** We derive a generalization bound showing that even imperfect generative models can improve CATE estimation by selectively augmenting subsets of the covariate space. This not only reduces covariate shift but also increases the effective sample size, offering significant advantages in small-sample regimes. ③ **Empirical validation:** Through experiments on three datasets, we demonstrate that GATE improves the performance of a wide range of CATE models compared to the no-augmentation baseline, while reducing the performance gap between learners. We further explore the benefits and drawbacks of instantiating our framework with LLMs, showing that LLMs' ability to utilise external knowledge offers particular benefits in low-data regimes.

## 2 PROBLEM SETUP

**Conditional average treatment effects (CATEs).** We assume access to an observational dataset $\mathcal{D}^{(\mathrm{obs})} = \{(X_i, T_i, Y_i)\}_{i=1}^n$ such that $(X_i, T_i, Y_i) \overset{\text{i.i.d.}}{\sim} P(X, T, Y)$, where $Y_i \in \mathcal{Y}$ is a continuous or binary outcome, $X_i \in \mathcal{X} \subset \mathbb{R}^d$ is a vector of covariates and $T_i \in \{0, 1\}$ is a binary treatment assignment. For conciseness, we ignore the sample subscript $i$ unless explicitly needed. Following the potential outcomes framework (Rubin, 1974), we assume that there are two possible *potential outcomes*: $Y(0)$ (no treatment) and $Y(1)$ (under treatment). Our overall goal is to estimate CATE:

$$\tau(x) = \mathbb{E}[Y(1) - Y(0) \mid X = x] = \mu_1(x) - \mu_0(x), \qquad (1)$$

where $\mu_t(x) = \mathbb{E}[Y(t) \mid X = x]$. We make the standard (Rubin, 1974) assumptions of *overlap* ($0 < \mathbb{P}(T = 1 \mid X = x) < 1 \; \forall x \in \mathcal{X}$), *ignorability* ($(Y(1), Y(0)) \perp T \mid X$), and *consistency* ($Y = Y(t)$ if $T = t$). We also define $P_t = P(X \mid T = t)$ and $\pi_t = \mathbb{P}(T = t)$.

**Conditional Average Potential Outcomes (CAPOs).** A problem closely related to CATE estimation is the estimation of the CAPO functions, $\mu_0(x)$ and $\mu_1(x)$. Although estimating $\mu_0$ and $\mu_1$ is not

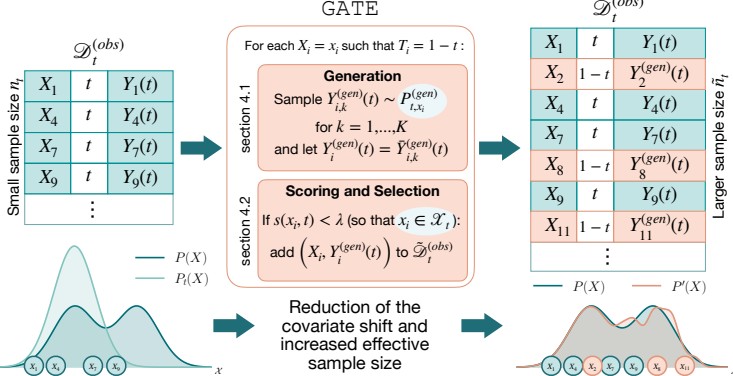

Figure 1: **Method overview.** GATE enhances CATE estimation in the finite-sample regime through selective data augmentation. For a fixed treatment $t$, synthetic potential outcomes for samples with $T_i = 1 - t$ are generated using $P_{t,x}^{(\text{gen})}$ and scored via $s(x,t)$ to decide their inclusion in $\tilde{\mathcal{D}}_t^{(\text{obs})}$. This approach mitigates covariate shift while boosting the effective sample size.

strictly necessary to obtain the treatment effect, it is sufficient and almost all the existing meta-learning strategies for estimating CATE (Künzel et al., 2019; Curth & Van der Schaar, 2021b) estimate $\mu_0$ and $\mu_1$ as one of the intermediate steps. For this reason we will focus our analysis on the estimation of $\mu_0$ and $\mu_1$ from $\mathcal{D}_0^{(\text{obs})}$ and $\mathcal{D}_1^{(\text{obs})}$ respectively, where $\mathcal{D}_t^{(\text{obs})} = \{(X_i, Y_i) \in \mathcal{D}^{(\text{obs})} \mid T_i = t\}$.

**The challenges of estimating CAPO from finite data.** Inferring CAPOs from finite observational data $\mathcal{D}_t^{(\text{obs})}$ is challenging for two main reasons (see Johansson et al. (2022) for a detailed discussion):

① **Large variance:** As in supervised learning, model variability will be large in *small datasets*, leading to less reliable predictions (Hastie et al., 2009). In context of CAPO, this issue is exacerbated because $\mathcal{D}_t^{(\text{obs})}$ may be significantly smaller than $\mathcal{D}^{(\text{obs})}$, since $\min\{|\mathcal{D}_0^{(\text{obs})}|, |\mathcal{D}_1^{(\text{obs})}|\} \leq \frac{1}{2}|\mathcal{D}^{(\text{obs})}|$, leading to relatively high error in CAPO estimation.

② **Covariate shift:** In observational data, covariates in the subset $\mathcal{D}_t^{(\text{obs})}$ are sampled from $P_t(X)$, which is typically different from $P_{1-t}(X)$ and hence also different from the marginal $P(X)$. As a result, a CAPO model fitted on $\mathcal{D}_t^{(\text{obs})}$ might not generalise well to the entire population.

## 3 GATE: DATA AUGMENTATION FOR IMPROVED PO ESTIMATION

Our objective in this work is to simultaneously minimise the covariate shift *and* decrease the model variability. Both of these goals could be achieved by generating the missing potential outcomes for individuals in $\mathcal{D}^{(\text{obs})}$ using **a generative model** $P_{t,x}^{(gen)}$ which allows to sample $Y^{(\text{gen})}(t) \mid X = x \sim P_{t,x}^{(gen)}$ for all $x \in \mathcal{X}$. Using this model, for every $(X, T, Y)$ we can create an additional sample $(X, 1 - T, Y^{(\text{gen})}(1 - T))$ and add it to $\mathcal{D}^{(\text{obs})}$ to create the augmented dataset $\tilde{\mathcal{D}}^{(\text{obs})}$ (see Figure 1).

If the generator $P_{t,x}^{(\text{gen})}$ was perfect, augmenting $\mathcal{D}^{(\text{obs})}$ by generating the missing potential outcomes for *all* individuals would fix the covariate shift problem. Indeed, the augmented dataset $\tilde{\mathcal{D}}_t^{(\text{obs})}$ would comprise the same covariates as $\tilde{\mathcal{D}}_{1-t}^{(\text{obs})}$. Furthermore, we would have $|\tilde{\mathcal{D}}_t^{(\text{obs})}| = n \gg \pi_t \cdot n$, hence mitigating the variance problem. However, in practice, the generator $P_{t,x}^{(\text{gen})}$ may be inaccurate (e.g. non-null bias), at least in some areas of the covariate space $\mathcal{X}$. In this case, to balance the bias introduced by $P_{t,x}^{(\text{gen})}$ with the benefits obtained by mitigating the covariate shift, it might be better to generate the missing potential outcomes only in a selective subset of the covariate space. We verify this intuition theoretically by deriving a generalisation bound.

### 3.1 BOUNDING THE GENERALIZATION ERROR OF THE AUGMENTED DATASET.

We fix a treatment $t \in \{0, 1\}$ and focus on augmenting the dataset $\mathcal{D}_t^{(\text{obs})}$, used to estimate $\mu_t$. To avoid introducing too much bias, we would like to use $P_{t,x}^{(\text{gen})}$ only in a selected subset of the covariate

space, which we call the *admissible set* $\mathcal{X}_t$. In what follows, we theoretically analyse how the choice of $\mathcal{X}_t$ affects the trade-off described in the previous section.

We note that by augmenting the dataset $\mathcal{D}^{(\text{obs})}$, we manipulate both the distribution of the covariates and the conditional distribution of the outcomes. To formalise the induced changes, we introduce the variables $(A, X', Y'(t))$, where $A$ is a binary random variable which indicates whether $X'$ is drawn from the factual distribution $P_t$ or not. That is, let $X'|A = t \sim P_t$ (so that $X'$ is drawn from the factual distribution when $A = t$) and let $X'|A = 1 - t \sim Q$, so that for $A = 1 - t$ we draw $X'$ from a distribution $Q$, supported on $\mathcal{X}_t$ only. Setting $\mathbb{P}(A = t) = \pi_t$, we obtain the following marginal distribution of $X'$: $P'(X') = \pi_t P(X' \mid T = t) + (1 - \pi_t)Q(X')$. We then let $Y'(t) = \mathbb{1}(A = t)Y(t) + \mathbb{1}(A = 1 - t)Y^{(\text{gen})}(t)$, i.e. we use the generative model only when $X'$ is drawn from $Q$, else we use the observed outcome.

Having formalized the data augmentation process with the distribution $Q$, we now highlight the trade-off associated with choosing a good $\mathcal{X}_t$ (and a good $Q$ supported on this $\mathcal{X}_t$) for CAPO estimation. We do so by deriving a generalization bound on the expected risk $R(f_t) = \mathbb{E}_{X,Y(t)}\left[L(Y(t), f_t(X))\right]$ for a hypothesis $f_t$ and a loss function $L$, building on the bound presented in (Johansson et al., 2022) to account for the effect of using the augmented dataset $\tilde{\mathcal{D}}_t^{(\text{obs})} = \{(X_i', Y_i'(t))\}_{i=1}^{\tilde{n}_t}$.

**Theorem 3.1.** *Assume access to an augmented dataset $\{(X_i', Y_i'(t))\}_{i=1}^{\tilde{n}_t} \overset{i.i.d.}{\sim} P'(X', Y'(t))$. Then with probability at least $1 - \delta$,*

$$R(f_t) \leq \tilde{R}^{(\text{emp})}(f_t) + (1 - \pi_t)IPM_{\mathcal{L}}\left(Q, P_{1-t}\right) + V_{P'}\frac{\mathcal{C}_{\tilde{n}_t, \delta}^{\mathcal{H}}}{\tilde{n}_t^{3/8}} \tag{2}$$

$$+ (1 - \pi_t)\mathbb{E}_{X \sim Q}\left[IPM_{\mathcal{L}^x}\left(P(Y(t) \mid X), P^{(gen)}(Y^{(gen)}(t) \mid X)\right)\right], \tag{3}$$

*where $\tilde{R}^{(\text{emp})}(f_t)$ is the empirical risk, $IPM_{\mathcal{S}}\left(P, P'\right) = \sup_{f \in \mathcal{S}}|\mathbb{E}_{V \sim P}[f(V)] - \mathbb{E}_{W \sim P'}[f(W)]|$ is the Integral Probability Metric between $P$ and $P'$, $\mathcal{L}$ and $\mathcal{L}^x$ are classes of functions determined by the choice of the loss $L$, $V_{P'}$ is a constant, and $\mathcal{C}_{\tilde{n}_t, \delta}^{\mathcal{H}} = \mathcal{O}(\log \tilde{n}_t^{3/8})$ as $\tilde{n}_t \to +\infty$.*

*Proof.* The proof is given in Appendix C, with a detailed definition of all the terms involved. $\square$

**Interpretation.** The bound in Theorem 3.1 illustrates the different mechanisms which can affect the generalization error involved in using generative models for data augmentation.

① *Covariate shift and Variance* : The term $IPM_{\mathcal{L}}\left(Q, P_{1-t}\right)$ measures the distance between the distribution $Q$ of the covariates for which the generative model is used and the counterfactual distribution $P_{1-t}$. On top of that, the term $V_{P'}\frac{\mathcal{C}_{\tilde{n}_t, \delta}^{\mathcal{H}}}{\tilde{n}_t^{3/8}}$ quantifies the variance stemming from the finite-sample regime, emphasising that potential outcome generation is particularly impactful in the small-sample regime. Both of these terms can be made small by performing data augmentation for *all* the samples with treatment $T = 1 - t$ in $\mathcal{D}^{(\text{obs})}$.

② *Noise of the generator* : While the minimization of the above two terms suggests that we should augment the observational dataset with as many generated samples as possible, this ignores the impact of the inaccuracy of the generator. This inaccuracy is highlighted by the term involving $IPM_{\mathcal{L}^x}$ in the bound, which quantifies how close the distribution $Y^{(\text{gen})}(t)$ is to the ground-truth distribution of the potential outcome $Y(t)$, conditioned on different values of the covariates $X$.

**Remark.** Recent work by Aloui et al. (2023) has similarly explored the role of data augmentation in CATE estimation, highlighting the fundamental tradeoff between reducing covariate shift and controlling imputation error. Our theoretical bound, while covering similar phenomena, offers several novelties. First, it provides a more general formulation by accommodating an arbitrary loss function $L$. Second, our bound explicitly accounts for the potential stochasticity of generative models, employing an IPM term to measure the distributional distance between the ground-truth outcome and the generator's distribution. Finally, our bound characterizes the finite-sample regime, showing how the benefits of data augmentation correlate with sample size.

**Navigating the trade-off through the admissible set $\mathcal{X}_t$.** The derived generalisation bound confirms that performance gains could be obtained even in the face of potential inaccuracy of $P_{t,x}^{(\text{gen})}$, by

balancing the effect of these two mechanisms with a careful choice of $Q$. We would like $Q$ to be as close as possible to the counterfactual distribution $P_{1-t}$ (to minimise term $\textcircled{1}$), while at the same time excluding regions where the generative model is particularly inaccurate (thus minimising term $\textcircled{2}$). To achieve this, for a given admissible set $\mathcal{X}_t$ we define $Q(X) \propto P_{1-t}(X)\mathbb{1}\{X \in \mathcal{X}_t\}$. The corresponding empirical distribution is then $\hat{Q}(x) = \frac{1}{|\mathcal{K}|} \sum_{i \in \mathcal{K}} \delta(x - X_i)$, with $\delta(x)$ the Dirac delta function and $\mathcal{K} = \{i \mid i \in [n], T_i = 1 - t, X_i \in \mathcal{X}_t\}$. This way, manipulating $\mathcal{X}_t$ for a given generative model $P_{t,x}^{(\text{gen})}$ allows to navigate the trade-off between the bias introduced by the inaccuracy of the generator, and the reduction of variance and covariate shift achieved via data augmentation. We provide a discussion of how $\mathcal{X}_t$ can be defined in Section 4.2.

# 4 INSTANTIATING GATE WITH LARGE LANGUAGE MODELS

As we discussed in the section above, the design of GATE involves making two design choices which will determine the efficacy of data augmentation: the choice of the generative model and the choice of the admissible set $\mathcal{X}_t$. In what follows, we describe an instantiation of GATE using LLMs, in which we propose to define the admissible set $\mathcal{X}_t$ using the LLM's uncertainty.

## 4.1 LARGE LANGUAGE MODELS AS POTENTIAL OUTCOME GENERATORS

The efficacy of the GATE framework hinges on the selection of the generative model $P_{t,x}^{(\text{gen})}$. By choosing $P_{t,x}^{(\text{gen})}$ that closely approximates the true conditional distribution $P(Y(t)|X = x)$, we can tighten the generalization bound presented in Theorem 3.1.

While GATE can be be instantiated with generative models trained exclusively on observational data $\mathcal{D}_t^{(\text{obs})}$ – such as local regression models (Aloui et al., 2023)) – the utility of such solutions is inherently limited since such generative models ultimately utilise the same information as the downstream CATE model. To overcome this limitation, we propose to instantiate GATE with foundational models, such as large language models (LLMs) (Bommasani et al., 2021).

**Why LLMs?** LLMs bring several unique properties that make them particularly well-suited for this task. Due to extensive pretraining they encode rich domain knowledge. As such, they can effectively utilize dataset metadata, such as descriptions of covariates, study populations, or the context of data collection, to align their outputs with the specific problem domain and integrate contextual relationships that may not be apparent in $\mathcal{D}_t^{(\text{obs})}$. This is particularly advantageous in small-sample regimes. Additionally, LLMs excel at few-shot learning, allowing them to adapt to a given task when conditioned on subsets of $\mathcal{D}_t^{(\text{obs})}$. By grounding their predictions in observed data while incorporating broader priors, LLMs can address shortcomings of $\mathcal{D}^{(\text{obs})}$ and improve alignment with real-world distributions, thus allowing the downstream CATE models to *transcendent observational data*.

To harness the capabilities of LLMs, we employ two complementary prompting strategies, which – as we show in our empirical experiments (Section 6) – allow GATE to significantly improve CATE estimators, particularly in data-scarce settings. Details of the prompts can be found in Appendix E.

**(I) Metadata-Driven Prompts.** We guide the extraction of the prior knowledge of the LLM by utilising metadata and auxiliary information present in observational datasets. We achieve this by including in the prompts information such as: natural language descriptions of covariates, information about the data collection technique, the population of the study or more general context of the dataset.

**(II) Conditioning the LLM on the observational dataset.** We exploit the few-shot learning capabilities of the LLM by conditioning the generation on a randomly chosen subset of samples from the observational dataset $\mathcal{D}^{(\text{obs})}$, presented in natural language, hence exploiting its in-context learning abilities.

**Stochastic nature of the LLM.** Given the stochastic nature of LLMs, for each sample $X_i = x_i$ in $\mathcal{D}^{(\text{obs})}$ for which we decide to generate the potential outcome, we propose to sample $K$ potential outcomes from $P_{t,x}^{(\text{gen})}$: $Y_{i,k}^{(gen)}(t) \sim P_{t,x_i}^{(gen)}, k = 1, \dots, K$. To improve the robustness of the generation, we then aggregate these samples by taking the mean, and setting $Y_i^{(gen)}(t) = \bar{Y}_{i,k}^{(gen)}(t)$ (see Figure 1). As we explain in the next section, we further use the variance in the generated potential outcomes to guide the selection of the admissible set $\mathcal{X}_t$.

## 4.2 CHOOSING THE ADMISSIBLE SET $\mathcal{X}_t$

The definition of the subset $\mathcal{X}_t$ for the selection of the generated outcomes $Y^{(gen)}$ is another important component of GATE. Under our definition of $Q$, choosing $\mathcal{X}_t = \emptyset$ retains the original observational dataset, failing to address the covariate shift. Conversely, setting $\mathcal{X}_t = \mathcal{X}$ mitigates covariate shift, but can introduce significant bias if the generated potential outcomes $Y^{(gen)}$ are imperfect. As motivated in Theorem 3.1 by the term ②, the choice of $\mathcal{X}_t$ can be guided by the objective of excluding from the admissible set regions of the covariate space $\mathcal{X}$ where the distribution $P_{t,x}^{(gen)}$ significantly deviates from the true distribution $P(Y(t)|X = x)$.

However, assessing the statistical distance between these two distributions is challenging, particularly when the available observational dataset is small, because the potential outcome $Y(t)$ is not observed for every $X = x$. Given this, we consider a scoring function $s(x, t)$, which is chosen as a *proxy* for the fidelity of $P_{t,x}^{(gen)}$ at a given point $x \in \mathcal{X}$. Then, for a fixed parameter $\alpha \in [0, 1]$, we define an adaptive threshold $\lambda(\alpha, \mathcal{D}^{(obs)}) = \text{Quantile}_\alpha(\{s(X_i, T_i) \mid i \in [n]\})$. [2] With this threshold, we define the admissible sets $\mathcal{X}_0$ and $\mathcal{X}_1$ as

$$\mathcal{X}_0 = \mathcal{X}_1 = \{X_i \mid i \in [n], s(X_i, T_i) < \lambda(\alpha, \mathcal{D}^{(obs)})\}, \tag{4}$$

comprising the samples for which the fidelity of the potential outcome generated by the $P_{t,x}^{(gen)}$ is below the $\alpha$-quantile. We provide a detailed discussion on this definition in Appendix B.3. Furthermore, we explore the influence of $\alpha$ on the performance of the downstream CATE model in our empirical experiments in Section 6.

**Choosing the scoring function $s$ for LLMs.** For deterministic models trained on $\mathcal{D}^{(obs)}$, properties of $\mathcal{D}^{(obs)}$ itself could be leveraged to find an optimal admissible set (Aloui et al., 2023; Jesson et al., 2020). For stochastic models such as LLMs, trained on data different from $\mathcal{D}^{(obs)}$, we propose to rely on the variance in the generated outcomes to define the admissible set, by setting $s(x, t) := \text{Var}_{Y^{(gen)}(t) \sim P_{t,x}^{(gen)}}(Y^{(gen)}(t) \mid X = x)$. While this scoring function might not always be optimal, it reflects the heuristic that the accuracy of the $P_{t,x}^{(gen)}$ might be lower in the areas where the generative model is less certain about its predictions. Indeed, in the context of LLMs, it has been shown that uncertainty measures such as variance can be used to discriminate between factually correct and incorrect responses (Huang et al., 2023; Manakul et al., 2023), as well as predict the quality of a response (Lin et al., 2023). A detailed discussion and evaluation of the variance-based scoring function can be found in Appendix B.3.

## 5 RELATED WORKS

**CATE Meta-Learners.** Model-agnostic approaches to CATE estimation, known as meta-learners, are widely studied for their competitive performance and strong theoretical foundations (Künzel et al., 2019; Kennedy, 2023). Among these, *two-step learners* (Kennedy et al., 2017; Nie & Wager, 2021; Curth & Van der Schaar, 2021b) implicitly leverage data augmentation by estimating *all* missing potential outcomes in the first step. Unlike two-step learners, GATE *selectively* imputes outcomes based on the generative model's properties. This flexibility allows GATE to act as a pre-processing step in the meta-learning pipeline, offering performance gains (e.g., by regularization or incorporating external knowledge) without altering the standard meta-learners. We empirically demonstrate GATE's benefits also to two-step learners in Section 6, with further discussion in Appendix A.

**Data Augmentation for CATE Estimation.** Few works address augmenting datasets with missing potential outcomes beyond the two-step learner framework. GANITE (Yoon et al., 2018) uses GANs to generate proxies for missing outcomes, while Aloui et al. (2023) selectively impute outcomes with local regression models. Unlike GATE, these methods do not provide a principled way to utilize generative models trained on *external datasets*, which is critical in small-sample or high covariate shift settings. We compare GATE against GANITE and COCOA in Section 6.3 and Appendix F.1, showing improved performance across datasets.

---

[2]Here, we propose a percentile-based threshold $\lambda$ to easily control the proportion of generated potential outcomes across datasets. However, in real-world applications, a fixed-value threshold (informed by domain knowledge) may better safeguard against low-quality outcomes. See Appendix B.3 for details.

# 6 NUMERICAL EXPERIMENTS

## 6.1 EXPERIMENTAL DESIGN

**Data.** Evaluating CATE models using observational data is challenging due to the lack of ground-truth CATE values. Standard benchmarks like IHDP (Hill, 2011) or News (Johansson et al., 2016) address this by designing artificial potential outcome functions. However, since we aim to compare generative models trained on $\mathcal{D}^{(\mathrm{obs})}$ with those trained on external (real-world) datasets, the outcome's relationship with treatment and covariates must be *reality-grounded*. Consequently, we utilize the following datasets: **Lalonde CPS1** (LaLonde, 1986) ($n = 7279$), where synthetic counterfactual outcomes are obtained using a pre-trained generative model from RealCause (Neal et al., 2020); **STAR Project** (Achilles et al., 2008) ($n = 1429$), obtained by subsampling an RCT dataset using the method from (Gentzel et al., 2021) to induce covariate shift, with 'ground-truth' CATE estimates derived from the full RCT; and **Hillstrom** (Hillstrom, 2008) ($n = 9639$), obtained similarly. Additional experiments on the IHDP dataset are in Appendix F.2. We partition these datasets into training $\mathcal{D}^{(\mathrm{obs})}$ and test $\mathcal{D}_{\mathrm{test}}$ sets of equal sizes, and standardize both covariates and continuous outcomes before inputting them into the CATE learners.

**CATE Models.** We evaluate GATE by comparing the performance of downstream CATE models when trained on the original dataset $\mathcal{D}^{(\mathrm{obs})}$ vs. when trained on the augmented $\tilde{\mathcal{D}}^{(\mathrm{obs})}$. From the portfolio of CATE meta-learners, we use the S-, T-, X-, R-, IPW- and DR-learner (Künzel et al., 2019; Curth & Van der Schaar, 2021b). We also consider the CFR-Wass and CFR-MMD algorithms (Shalit et al., 2017) relying on balanced representations, designed specifically to tackle the covariate shift (Johansson et al., 2022). We further complete the list of considered models with TARNet (Shalit et al., 2017), DragonNet (Shi et al., 2019) and BART (Athey & Imbens, 2016).

**Instantiating GATE.** To test the potential of instantiating GATE with LLMs, we use GPT-3.5 Turbo (Achiam et al., 2023) – a widely used and reliable model, well-suited for robust evaluations of our framework. Further, we define the admissible set $\mathcal{X}_t$ with the variance-based criterion (Section 4.2), using a fixed threshold $\alpha = 0.5$ unless otherwise stated (note that this value is *not* tuned between datasets or experiments, to allow for fair evaluation). We also use 100 in-context samples in each prompt unless otherwise stated. In experiments in section Section 6.3, we compare the performance of the LLM-instantiated GATE against a diverse set of models trained on $\mathcal{D}^{(\mathrm{obs})}$, which include: mean model, 1-nearest neighbour (1-NN), random forest (RF) and GAN (following the approach of GANITE (Yoon et al., 2018)) (detailed description of these models can be found in Appendix D.3).

**Takeaways.** The takeaways from the experiments are organised as follows:
▶ *Section 6.2:* LLM-instantiated GATE improves performance across diverse datasets and benchmark meta-learners.
▶ *Section 6.3:* Although multiple generative models used within GATE provide performance benefits for CATE estimation, leveraging LLMs provides substantial performance gains over models trained solely on $\mathcal{D}^{(\mathrm{obs})}$, particularly in low-sample regimes.
▶ *Section 6.4:* Prompting the LLM with meta-data and contextual information is crucial to elicit its prior knowledge. In-context samples further allow to improve its performance.
▶ *Section 6.5.1:* GATE is particularly advantageous in problems with strong covariate shift.
▶ *Section 6.5.2:* Adjusting the admissible set $\mathcal{X}_t$ through the hyperparameter $\alpha$ enables balancing the bias introduced by $P_{t,x}^{(\mathrm{gen})}$ against the reduction in covariate shift, optimizing overall performance.

More detailed descriptions of the experiments can be found in Appendix D. Anonymised code to reproduce the experiments can be found here.

## 6.2 DOES DATA AUGMENTATION WITH GATE IMPROVE CATE ESTIMATION?

**Goal.** In this experiment, we verify that LLM-instantiated GATE consistently improves the performance of different CATE meta-learners, across all benchmark datasets.

**Setup.** Each CATE model is trained on both the original dataset and the GATE-augmented dataset, and we compare the PEHE obtained in each case. The *only* difference between these two settings comes from the data used to train the CATE estimators (additional parameters such as architecture or hyperparameters were kept fixed between the two settings).

Table 1: **GATE performance with various CATE learners.** GATE improves the performance of different CATE learners across the datasets first without data augmentation (✗), and then with data augmentation (✓). Average $\sqrt{\epsilon_{\text{PEHE}}}$ and $1\text{std}$ is reported for 3 seeds (↓ is better)

| Learner | Lalonde CPS1D | | STAR | | Hillstrom | |
|---|---|---|---|---|---|---|
| | ✗ | ✓ | ✗ | ✓ | ✗ | ✓ |
| S-learner | 1.09±0.07 | 0.95±0.01 | 0.78±0.10 | 0.56±0.02 | 0.32±0.03 | 0.25±0.01 |
| T-learner | 1.28±0.03 | 0.96±0.01 | 0.81±0.08 | 0.50±0.03 | 0.4 ± 0.01 | 0.24±0.01 |
| X-learner | 1.43±0.10 | 0.95±0.01 | 0.93±0.05 | 0.49±0.02 | 0.29±0.01 | 0.24±0.01 |
| R-learner. | 1.35±0.42 | 0.95±0.00 | 6.12±2.57 | 0.47±0.01 | 0.63±0.21 | 0.26±0.02 |
| IPW-learner. | 1.12±0.03 | 0.95±0.01 | 0.57±0.06 | 0.47±0.01 | 0.29±0.01 | 0.25±0.00 |
| DR-learner | 1.29±0.02 | 0.95±0.01 | 0.60±0.11 | 0.48±0.02 | 0.41±0.02 | 0.25±0.01 |
| CFR-Wass. | 0.99±0.03 | 0.95±0.02 | 0.61±0.15 | 0.41±0.01 | 0.24 ± 0.0 | 0.24 ± 0.0 |
| CFR-MMD | 1.00±0.03 | 0.95±0.00 | 0.64±0.16 | 0.44±0.00 | 0.24±0.00 | 0.24±0.00 |
| TARNet | 1.20±0.03 | 0.96±0.01 | 0.49 ± 0.1 | 0.48±0.04 | 0.39±0.02 | 0.24±0.00 |
| DragonNet | 0.97±0.02 | 0.95±0.02 | 0.90±0.26 | 0.48±0.04 | 0.41±0.04 | 0.24±0.01 |
| BART | 1.36±0.03 | 1.35±0.00 | 0.70±0.09 | 0.56±0.02 | 0.27±0.02 | 0.25±0.01 |

**Results.** Table 1 shows that GATE consistently improves performance across all the considered CATE models, with gains across the average PEHE (Hill, 2011) and its standard deviation. Furthermore, GATE decreases the performance gap across CATE learners, making it a *model agnostic* data pre-processing step that can aid model selection, usable with both one-step and two-step learners.

### 6.3 WITHIN GATE, HOW DOES THE LLM COMPARE TO OTHER GENERATIVE MODELS?

**Goal.** To explore *whether* and *when* using the LLM proves particularly beneficial, we now compare the performance obtained with the LLM against alternative generative models.

**Setup.** We perform this comparison using the two-step DR-learner. For the baseline models described in Section 6.1, we train each $P_{t,x}^{(\text{gen})}$ on the respective $\mathcal{D}_t^{(\text{obs})}$. For the LLM, we generate $K = 10$ surrogate outcomes per sample, which we then average to obtain $Y^{(\text{gen})}$. Performance comparisons are conducted across three datasets of varying sizes, randomly sampled with proportions $\rho \in \{0.1, 0.5, 1.0\}$ from the original observational datasets. Each baseline uses the same admissible sets $\mathcal{X}_t$ as the LLM for a fair comparison. Results with $\mathcal{X}_t = \mathcal{X}$ can be found in Appendix F.4.

**Results.** In Figure 3, we present the average $\sqrt{\epsilon_{\text{PEHE}}}$ obtained across 3 seeds when instantiating GATE with each of the generative models. We note that multiple models can offer performance improvements to the downstream CATE model, compared to the no-augmentation baseline. Interestingly, the LLM consistently outperforms the baselines trained on $\mathcal{D}^{(\text{obs})}$ only, yielding lower average PEHE. As predicted, this performance gap is most evident in the small sample regime ($\rho = 0.1$), where LLM-derived prior knowledge proves most beneficial. Remarkably, comparing the PEHE obtained across all three dataset sizes, our results demonstrate that using GATE with the LLM allows to obtain performance levels which are close to optimal when using only a fraction of the original dataset. The performance gap between LLMs and other generative models is most pronounced in the STAR dataset, which exhibits high treatment effect heterogeneity. Conversely, this gap is minimal in the Hillstrom dataset, where potential outcome heterogeneity is low (cf. Appendix F.4). This low heterogeneity explains the strong performance of the mean imputation model, as the generated *constant* potential outcomes effectively regularize the downstream model.

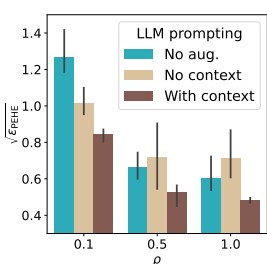

Figure 2: **Prior knowledge in GATE.** Contextual information helps to achieve good performance in low-samples. The error bars mark $1\text{std}$ (3 seeds).

### 6.4 WHERE DOES THE BENEFIT OF USING THE LLM COME FROM?

**Goal.** The surprisingly superior performance of the LLM over the non-parametric baselines can possibly be explained by two orthogonal factors: its in-context learning abilities, and its contextual understanding of the meta-data associated with the datasets. In this experiment, we disentangle these two components.

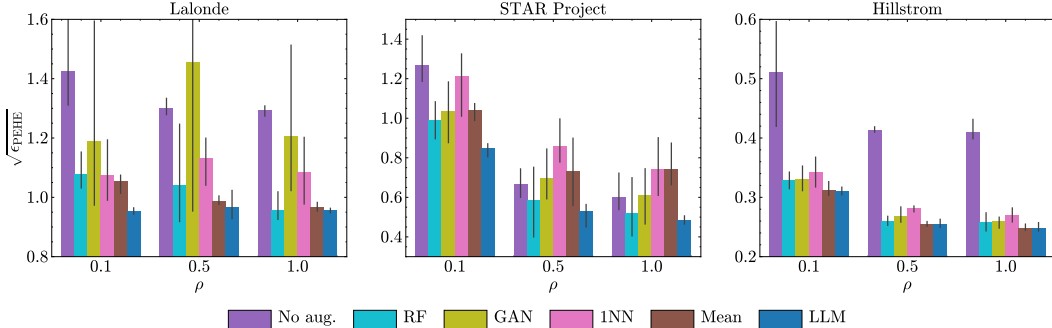

Figure 3: **Comparison of generative models in GATE.** Although multiple generative models offer performance improvements over the no-augmentation baseline, the LLM outperforms the models trained on $\mathcal{D}^{(\text{obs})}$ across different proportions $\rho$. The error bars mark $1\text{std}$, computed across 3 seeds.

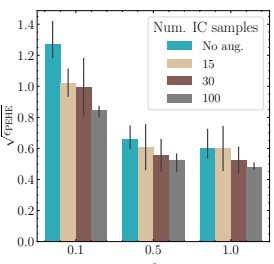

Figure 4: **In-context learning in GATE.** LLMs effectively learn from the provided in-context samples. The error bars mark $1\text{std}$ (3 seeds).

**Setup.** To quantify the influence of prior knowledge, we perform an ablation where we remove all the contextual information in the prompt given to the LLM and give the features generic names (e.g. *Feature 1*). Hence effectively only the in-context samples are provided. We compare the performance of this context-deprived LLM with an LLM which is informed about the context of the dataset and feature names (see Appendix E for exact prompts). Further, to evaluate the influence of the in-context samples provided, within the context-informed prompt we vary the number of samples included between $\{15, 30, 100\}$. We perform both experiments on the STAR dataset, with DR-learner as the learner and for varying proportions $\rho$.

**Results.** Figure 2 shows that substantial performance gains can be obtained when the meta-data of the STAR dataset is used to elicit the prior knowledge of the LLM, particularly when $\rho$ is small. As $\rho$ increases, the performance gap between context-informed LLM and the no-augmentation baseline naturally becomes smaller. Results for other datasets are in Appendix F.5. Furthermore, Figure 4 demonstrates that including more in-context samples in the prompt improves the downstream performance, showcasing LLM's ability to learn from the provided examples.

### 6.5 DOES GATE CONFORM TO THEORETICAL EXPECTATIONS?

Having shown that GATE can consistently improve CATE estimation, we now further verify whether the obtained empirical results agree with the theoretical results derived in Section 3.1. In particular, we investigate whether indeed using GATE allows to address the covariate shift problems, as well as whether the gains obtained from the covariate shift reduction can counterbalance the potential bias introduced by an imperfect generative model.

#### 6.5.1 IS GATE PARTICULARLY BENEFICIAL IN HIGH COVARIATE SHIFT SETTINGS?

**Goal.** We investigate the correlation between the performance gains obtained with GATE and the intensity of the *covariate shift* between the treated and control groups in $\mathcal{D}^{(\text{obs})}$.

**Setup.** We modulate the covariate shift's strength by adjusting the biasing intensity in the subsampling mechanism proposed by (Gentzel et al., 2021). This manipulation yields three distinct datasets, derived from the original STAR dataset. To quantify the strength of the covariate shift, we compute the sliced Wasserstein distance (SW) between the covariates of individuals in $\mathcal{D}_0^{(\text{obs})}$ and $\mathcal{D}_1^{(\text{obs})}$. For each dataset, we calculate the difference in $\sqrt{\epsilon_{\text{PEHE}}}$ obtained with CATE learners trained on $\tilde{\mathcal{D}}^{(\text{obs})}$ (obtained with the LLM) and $\mathcal{D}^{(\text{obs})}$.

**Results.** In Figure 5 (left), the performance gain obtained by GATE increases with the SW across almost all CATE models, validating the insight from Theorem 3.1. Contrary to other meta-learners, we find that the S-learner is least affected by data augmentation. We believe this to be due to its data efficiency (using all data for each PO estimate).

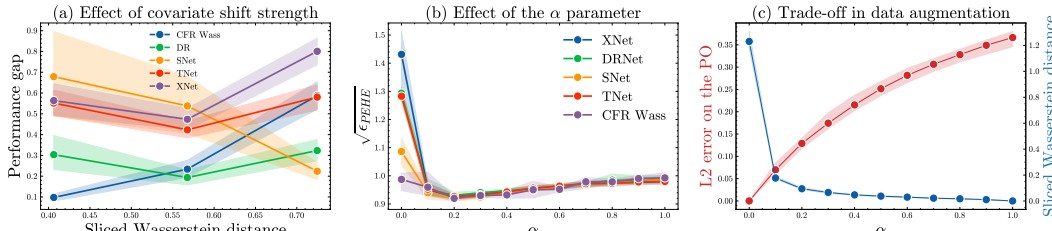

Figure 5: **Left:** The performance gains offered by GATE increase across the majority of learners as the strength of the covariate shift increases (the shaded regions denote 95% confidence intervals computed over 30 seeds) **Middle:** The value of the hyperparameter $\alpha$ allows to navigate the trade-off involved in data augmentation. **Right:** The bias introduced by $P_{t,x}^{(\text{gen})}$ is counterbalanced by the reduction in covariate shift obtained when using GATE.

### 6.5.2 CAN COVARIATE SHIFT REDUCTION COUNTERBALANCE THE BIAS INTRODUCED BY AN IMPERFECT GENERATIVE MODEL?

**Goal.** We further verify whether, as indicated by Theorem 3.1, the benefits obtained from the reduction in covariate shift can counterbalance the bias potentially introduced by $P_{t,x}^{(\text{gen})}$, thus offering performance benefits to the downstream CATE model. We also check whether the hyperparameter $\alpha$ allows to navigate the trade-off between the covariate shift reduction and the bias induced by $P_{t,x}^{(\text{gen})}$.

**Setup.** We vary the quantile value $\alpha$ used by the selector (Equation (4)) across the range $(0, 1)$. For each $\alpha$, we compute the performance when using GATE (with LLM) for the different CATE models (Figure 5, middle). Furthermore, we explicitly quantify the covariate shift in $\tilde{\mathcal{D}}^{(\text{obs})}$ using SW, and the bias introduced by $P_{t,x}^{(\text{gen})}$ by computing the average error in the potential outcomes in $\tilde{\mathcal{D}}^{(\text{obs})}$ compared to the ground-truth values, see Appendix D.5 for more details. We show how these quantities vary as we change $\alpha$ (Figure 5, right). We report averages and 95% confidence intervals for 3 seeds.

**Results.** Both the middle and right plots in Figure 5 verify our intuition that there exists an optimal choice of $\alpha$ ($\alpha \simeq 0.2$) for the Lalonde dataset which allows to balance the gains obtained by addressing the covariate shift with the losses suffered by introducing bias with $P_{t,x}^{(\text{gen})}$. Figure 5 (middle) also demonstrates that GATE offers performance gains for most choices of $\alpha > 0$ when compared to the no-augmentation baseline ($\alpha = 0$). Further, as we increase the allowed level of uncertainty of $P_{t,x}^{(gen)}$, the average $L^2$ error in potential outcomes over $\tilde{\mathcal{D}}^{(obs)}$ (quantifying how much noise is introduced by data augmentation) increases, while the covariate shift decreases. Despite this bias, optimal performance is obtained when $\alpha > 0$, as suggested by Theorem 3.1.

## 7 DISCUSSION AND FUTURE WORKS

**Model Selection.** Within the proposed implementation of GATE (see Section 3.1), using GATE requires making two key choices: selecting the generative model and choosing the value of the hyperparameter $\alpha$. We provide principles to guide the selection of the generative model in Section 4.1 and Section 6.3. Once the generative model is selected, the value of $\alpha$ can be tuned using standard model selection approaches in CATE estimation, noting that setting $\alpha = 0$ corresponds to the 'no augmentation' baseline. This means that if the generative model is not sufficiently accurate and might degrade downstream performance, using GATE can recover the baseline performance. We also note that while incorporating GATE introduces one additional hyperparameter, as we show in Section 6.2, it can reduce the performance gap between different CATE learners, thus potentially simplifying the model selection process.

**Broader Impact.** In Section 6 we learn that GATE may enable practical adoption of CATE estimation in low-sample settings, possibly yielding a positive impact in fields where data is costly. Furthermore, GATE helps address problems such as covariate shift (particularly in low-sample regimes), further aiding the adoption of CATE inference in practice. However, extra care should be taken before relying on the LLM to guide decision-making in high-stakes domains.

**Reproducibility statement.** We provide all the details on the datasets and the implementation of baselines in Appendix D. Furthermore, we detail the prompts used by the LLM-instantiated GATE in Appendix E. Anonymised code to reproduce the experiments can be found here.

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

# A RELATED WORKS

**Methods for CATE estimation.** Machine learning methods for CATE estimation can be broadly divided into two categories: model-specific and model-agnostic methods. **Method-specific** approaches rely on adjusting specific machine learning methods to the treatment effect setting. This gives rise to solutions based on neural networks (Shalit et al., 2017; Shi et al., 2019; Johansson et al., 2016; Curth & Van der Schaar, 2021a), Gaussian processes (Alaa & Van Der Schaar, 2017) or random forests and regression trees (Athey & Imbens, 2016; Wager & Athey, 2018; Hahn et al., 2020; Hill, 2011).

In contrast, **model-agnostic** methods (so-called 'meta-learners' (Künzel et al., 2019; Curth & Van der Schaar, 2021b)) are general learning strategies which can be instantiated with any base learner (e.g., neural network, random forest). Within the model-agnostic strategies we can distinguish the *one-step learners*, which directly estimate the potential outcome surfaces, $\hat{\mu}_0$ and $\hat{\mu}_1$, and then obtain the CATE as: $\hat{\tau} = \hat{\mu}_1 - \hat{\mu}_0$. The alternative *two-step learners* (Kennedy et al., 2017; Nie & Wager, 2021; Curth & Van der Schaar, 2021b) implicitly rely on ideas from data imputation. In their first step, two-step learners obtain pseudo outcomes $\tilde{Y}_\phi$, which are "proximal" target treatment effect values, composed from nuisance parameters $\phi = (\pi, \mu_0, \mu_1)$ estimated from the given observational dataset $\mathcal{D}^{(\text{obs})}$ (or a subset of it). In the second step, the final CATE model is obtained by regressing the pseudo-outcomes $\tilde{Y}_\phi$ on the covariates $X$. The pseudo-outcomes can be obtained using strategies relying on propensity weighting (IPW-learner (Horvitz & Thompson, 1952)), regression-adjustment (X-learner (Künzel et al., 2019)) or both of these combined (DR-learner (Kennedy, 2023)). We note that our framework GATE is a strategy which is complementary to these standard two-step learners.

**How is GATE different from a standard two-step learner?**

1. **Admissible set:** Two-step learners require obtaining the missing potential outcomes for *all* individuals in the observational dataset. As we demonstrate, this might introduce excessive bias if the generative model $P_{t,x}^{(\text{gen})}$ is inaccurate. As a solution to this problem, in our framework we introduce the concept of an admissible set $\mathcal{X}_t$, which allows to navigate the trade-off between the reduction of the covariate shift and the introduced bias. In the case where $\mathcal{X}_t \neq \mathcal{X}$, GATE does not allow to explicitly obtain treatment effect proxies for all individuals in the dataset, making it different from a standard two-step learner.

2. **External information:** A standard two-step learning strategy does not allow to utilise external sources of information to inform the generation of the pseudo-outcomes, as the nuisance parameters $\phi$ estimated in the first step are fitted using the observational data $\mathcal{D}^{(\text{obs})}$ only. In contrast, a key defining characteristic of GATE is that it allows to infuse the downstream CATE estimator with external knowledge, by training the generative model $P_{t,x}^{(\text{gen})}$ on datasets different from $\mathcal{D}^{(\text{obs})}$.

3. **Complementary inductive biases:** Considering our method as a pre-processing data augmentation method allows to aggregate the inductive biases imposed by the generative model $P_{t,x}^{(\text{gen})}$ and the downstream CATE learner used on the augmented dataset, $\hat{\tau}$. Particularly in small sample regimes, when the observational dataset $\mathcal{D}^{(\text{obs})}$ does not contain sufficient information to confidently estimate the CATE function, combining the inductive biases imposed by different methods might be particularly beneficial. This is why in our experiments (Section 6) we fit two-step learners on top of the GATE-augmented dataset, demonstrating performance improvements.

As such, GATE can be used as the first step in the meta-learning pipeline, without requiring any change to the standard meta-learners.

**Data augmentation for CATE estimation.** Other works have proposed alternative model-specific instantiation of the two-step learning strategy, relying for example on obtaining the pseudo-outcome using a GAN model (Yoon et al., 2018), or local regression methods (Aloui et al., 2023). However, these imputation approaches are constrained by the amount of information present in the observational datasets. As a result, they are particularly vulnerable to scenarios with covariate shift, where there are significant differences between the distributions of the control and treated groups. Furthermore, these imputation methods (GAN and local regression) require large amounts of data to be accurate,

which contrasts the small-sample regime tackled in this work. On the other hand, `GATE` provides a principled way of leveraging models trained on external data sources, such as the LLMs, and thus is able to take advantage of the dataset metadata to set the context, leading to helpful data augmentation as shown in Section 6.2.

**LLMs as sources of prior knowledge for downstream tasks.** As large language models (LLMs) have increased in parameter count and training set size, it has become clear that they are able to act as knowledge bases, showing great performance across a variety of knowledge-retrieval tasks (Brown et al., 2020; Nori et al., 2023; Li et al., 2024; He et al., 2024). As such, LLMs have been proposed as tools for extracting prior knowledge about the world, which can be used to ground standard *data-driven* ML models in real-world contexts and encourage their outputs to be consistent with common-sense reasoning based on the meta-data (Choi et al., 2022). Relying on the inductive biases generated by the LLMs from the task-specific metadata has been demonstrated to improve performance on tasks as diverse as reinforcement learning (Choi et al., 2022; Du et al., 2023), tabular learning (Zhu et al., 2023; Seedat et al., 2024) as well as causal discovery (Choi et al., 2022; Takayama et al., 2024; Jiralerspong et al., 2024; Ban et al., 2023). Furthermore, recent studies demonstrate that LLMs are in principle capable of gaining knowledge about the underlying causal structure of real-world data generating processes, despite not being explicitly trained to reason 'causally' (Richens & Everitt, 2024; Zečević et al., 2023). This finding is further supported by empirical research, demonstrating LLMs potential for answering causal queries (Long et al., 2024; Willig et al., 2022; Zečević et al., 2023). This further motivates the use of LLMs as sources of prior knowledge and relevant inductive biases in *causal inference* tasks in particular.

**Comparison with domain adaption.** Our bound in Theorem 3.1 is related to a series of works studying generalisation theory for unsupervised domain adaptation (Ben-David et al., 2006; Long et al., 2015; Mansour et al., 2009), but differs in significant ways. These bounds involve the risk in the target domain ($D = 1$) using the observed risk in the source domain ($D = 0$) and the distance between the domains:

$$R_{D=1}(f) \leq R_{D=0}(f) + d_{\mathcal{H}}(P(X|D=1), P(X|D=0)) + \lambda_{\mathcal{H}},$$

where $\mathcal{H}$ is some function class and $\lambda_{\mathcal{H}}$ is a constant. Unique in our bound is the use of the distribution $Q$ to split $d_{\mathcal{H}}$ into the terms $IPM_{\mathcal{L}}(Q, P_{1-t})$ and $\mathbb{E}_{X \sim Q}[IPM_{\mathcal{L}^x}(P(Y(t)|X = x), P(Y^{(gen)}(t)|X = x))]$ which correspond respectively to the covariate shift and the bias introduced by the generator. Through this, our bound offers the following novel insights which we validate experimentally:

1. **Insight**: Even when the generative model $P_{t,x}^{(gen)}$ is imperfect, using it to target covariate shift can improve performance.

   → **Experiment 6.5.2**: We explicitly compare the bias introduced by the generative model $P_{t,x}^{(gen)}$ against the reduction of the covariate shift obtained by data augmentation, showing that these two effects can be balanced.
   → **Experiment 6.4**: We verify this by comparing the performance with and without `GATE` across three datasets and multiple CATE models.

2. **Insight**: Tuning the distribution $Q$ via the admissible set $\mathcal{X}_t$ allows to balance the trade-off between the bias introduced by the generator, and the reduction of variance and covariate shift achieved via data augmentation.

   → **Experiment 6.5.2**: We verify that modulating $\mathcal{X}_t$ allows to navigate this trade-off.

3. **Insight**: Excluding from the admissible set regions of the covariate space where the generative model is particularly "incorrect" can improve performance.

   → **Experiment 6.5.2**: We propose to identify such regions using a proxy measure: the uncertainty in the generated outcomes. We verify that as we increase the allowed level of uncertainty of $P_{t,x}^{(gen)}$, the bias introduced by data augmentation increases, while the covariate shift decreases.

# B   DETAILS ON GATE

## B.1   USAGE WITH CATE LEARNERS

GATE is a data augmentation method, which means that it is agnostic to the choice of the downstream CATE learner (Curth & Van der Schaar, 2021a; Künzel et al., 2019). As such, it can be used both with one-step learners and two-step learners. We illustrate in Algorithm 1 how to use it in practice.

---

**Algorithm 1** Using GATE with CATE meta-learners.

---

**Input**: observational dataset $\mathcal{D}^{(\text{obs})} = \{(X_i, T_i, Y_i)\}_{i=1}^{n}$, pretrained generative model $P_{t,x}^{(\text{gen})}$, admissible sets $\mathcal{X}_0, \mathcal{X}_1$
**Output**: CATE estimation model $\hat{\tau}(x)$

 1: **for** $t \in \{0, 1\}$ **do**
 2:     $\tilde{\mathcal{D}}_t^{(obs)} \leftarrow \mathcal{D}_t^{(obs)}$
 3:     **for** $i = 1$ to $n$ **do**
 4:         **if** $T_i = 1 - t$ and $X_i \in \mathcal{X}_t$ **then**
 5:             sample $Y^{(\text{gen})} \sim P_{t,X_i}^{(gen)}$
 6:             $\tilde{\mathcal{D}}_t^{(obs)} \leftarrow \tilde{\mathcal{D}}_t^{(obs)} \cup \{(X_i, Y^{(\text{gen})})\}$
 7:         **end if**
 8:     **end for**
 9: **end for**
10: **T-learner:** For $t = 0, 1$, fit $\hat{\mu}_t(x)$ on $\tilde{\mathcal{D}}_t^{(obs)}$, then $\hat{\tau}(x) = \hat{\mu}_1(x) - \hat{\mu}_0(x)$;
11: **S-learner:** Fit $\hat{\mu}(x, t)$ on $(\tilde{\mathcal{D}}_0^{(obs)}, \tilde{\mathcal{D}}_1^{(obs)})$, then $\hat{\tau}(x) = \hat{\mu}(x, 1) - \hat{\mu}(x, 0)$;
12: **Two-step learners:** For $t = 0, 1$, fit $\hat{\mu}_t(x)$ on $\tilde{\mathcal{D}}_t^{(obs)}$ and fit $\hat{\pi}(x)$ on $\tilde{\mathcal{D}}^{(\text{obs})}$; perform the second step on $\mathcal{D}^{(\text{obs})}$ or on a held-out observational dataset.

---

*One-step learners*: Examples of one-step learners include the T-learner and the S-learner. For the T-learner, one can estimate separately each $\mu_t$ using the dataset $\tilde{\mathcal{D}}_t^{(obs)}$. For the S-learner, we define the concatenation $\tilde{\mathcal{D}}^{(obs)} = (\tilde{\mathcal{D}}_0^{(obs)}, \tilde{\mathcal{D}}_1^{(obs)})$ which shall be used to estimate $\mu(x, t)$, the average PO for treatment $t$ and covariate $x$.

*Two-step learners*: Two-step learners require the estimation of the nuisance parameters $\mu_0$ and $\mu_1$ in their first step, which we propose to estimate on $\tilde{\mathcal{D}}^{(\text{obs})}$. In addition, some two-step learners (e.g. DR learner) require an estimation of the propensity score $\pi(x)$. Such estimator can be obtained by considering either the original dataset $\mathcal{D}^{(\text{obs})}$ or the augmented dataset $\tilde{\mathcal{D}}^{(\text{obs})}$ (in our empirical experiments, we used the latter option). The second step of these learners does not require any change as the nuisance estimators are used as plug-in. As such, the pseudo-outcomes should be obtained for the observational dataset $\mathcal{D}^{(obs)}$.

## B.2   THE QUESTION OF MODEL SELECTION AND HYPERPARAMETER TUNING

The fundamental problem of causal inference makes the standard approaches to model selection and hyperparameter tuning not applicable in CATE estimation. Because the ground truth CATE value is unobserved, one cannot simply choose a model which performs best on a held-out validation set. Instead, model selection for CATE estimation has to rely on heuristics, assumptions on the data generating process and general prior knowledge of the problem at hand (Curth & Van Der Schaar, 2023). Model selection procedures for causal inference models remain an active area of research (Saito & Yasui, 2020; Schuler et al., 2018; Lan & Syrgkanis, 2024; Curth & Van Der Schaar, 2023; Mahajan et al., 2022).

The challenges of model selection in causal inference also apply to GATE. Deciding which of the available generative models should be used to augment the observational dataset at hand is non-trivial, and neither is the question of choosing the admissible set $\mathcal{X}_t$ (e.g. by specifying the value of $\alpha$ in our proposed instantiation) for a given generative model.

**Choosing the generative model.** We provide the following insights which might guide the selection of the generative model within the GATE framework:

1. **LLMs vs other models.** The performance gap between the LLM and the models trained on $\mathcal{D}^{(\text{obs})}$ seems to depend on the size of the dataset (with LLMs providing particular performance improvements in smaller datasets, cf. Figure 3), as well as on the variability in the (standardized) potential outcomes (the Hillstrom dataset – where the performance gap is particularly small – has $Var(Y(1)) = 0.04$ and $Var(Y(0)) = 0.02$, while in the STAR dataset – where the performance gap is particularly large – $Var(Y(1)) = 1.05$ and $Var(Y(0)) = 0.93$). In scenarios with low outcome heterogeneity and/or large sample sizes, a simple model such as mean imputation can already perform well.

2. **Auditing the outcomes generated with the LLM**: Contrasting other generative models, the use of LLMs with GATE permits to make the generation process more transparent. Beyond producing numerical values, LLMs can also detail verbal explanations of their generations. Indeed, alternative prompting strategies can be employed to elicit explicit causal reasoning chains underpinning outcome generation. This capability enables human-in-the-loop applications of GATE, where domain experts can evaluate the generated data by examining these reasoning traces against their domain knowledge. As such, we view the use prompting techniques for explicit reasoning, such as chain-of-thought (Wei et al., 2022) or tree of thought(Yao et al., 2024), as a promising direction for future work.

**Tuning the hyperparameter $\alpha$.** With these challenges in mind, we propose three complementary strategies which allow to guide the selection of the value of $\alpha$ to define the admissible set $\mathcal{X}_t$ within the GATE framework:

1. As we explain above, relying on the **fixed-threshold definition of the admissible set** (rather than the percentile-based definition of the threshold) can allow to guide the selection of $\alpha$ using domain knowledge.

2. We further propose to guide the selection of $\alpha$ by **measuring the covariate shift** between the sets $\tilde{\mathcal{D}}_0^{(obs)}$ and $\tilde{\mathcal{D}}_1^{(obs)}$, using for example the sliced Wasserstein distance (an example of such an analysis can be found in our Figure 5, right). Then, we propose to choose the minimal value of $\alpha$ which allows to achieve significant reduction in a covariate shift (which might correspond to the 'elbow' in the graph). While such an 'elbow' might not always exist, this criterion provides additional guidelines in certain circumstances.

3. Finally, the choice of $\alpha$ can be further guided by **standard methods for CATE model selection**, particularly those based on comparing the downstream CATE models using a pseudo-outcome surrogate criteria evaluated on a held-out validation set (see Curth & Van Der Schaar (2023) for an overview). In particular, in view of strong covariate shift and small sample regime, we propose to rely on criteria which do not rely on estimating the propensity score, as these might lead to high variance in such cases.

Nevertheless, while finding the *optimal* value of $\alpha$ is non-trivial, our experiments showed the following: (1) Fixing $\alpha = 0.5$ consistently led to improved dowstream PEHE across the 3 datasets and the 11 CATE learners (cf. Table 1) (2) The strong effect of the reduction in covariate shift obtained with data augmentation (shown in Figure 5) reduces the sensitivity with respect to $\alpha$. Indeed, Figure 5 (middle) highlights that any value $\alpha > 0$ leads to performance gains compared to $\alpha = 0$.

### B.3 USING VARIANCE IN THE GENERATED OUTCOMES TO SELECT THE ADMISSIBLE SET

**Sources of variance.** We acknowledge that the variance in the outcomes generated by the LLM, which we use as a proxy to evaluate the LLM's uncertainty and hence guide the selection of the admissible set $\mathcal{X}_t$, might capture different types of uncertainty. Firstly, it might reflect the aleatoric uncertainty, which refers to the irreducible uncertainty of the outcome distributions. Secondly, it also captures the epistemic uncertainty, which accounts for both the insufficiency of observational data in some regions of the covariate space and insufficient semantic knowledge of the LLM. Our variance-based selection mechanism relies on the implicit assumption that the aleatoric uncertainty does not vary significantly across the covariate space $\mathcal{X}$. This implies that choosing the admissible set $\mathcal{X}_t$ based on the variance allows to capture the differences in the epistemic uncertainty of the LLM

across $\mathcal{X}$, where higher epistemic uncertainty may indicate to a higher inaccuracy in the generated outcomes.

**Definition of the admissible sets.** We define the admissible sets $\mathcal{X}_0$ and $\mathcal{X}_1$ in Section 4.2. In our instantiation, these sets are kept equal. The rationale for this choice is that only samples with relatively low uncertainty should be kept. One can imagine the situation where the generative model performs significantly worse for one of the groups (i.e. treated or control) compared to the other one. If the quantile value $\lambda(\alpha, \mathcal{D}^{(\mathrm{obs})})$ was computed separately for the treated and control groups, then the same ratio of samples would be kept in the augmented dataset in the two groups, despite the disparities across these groups. This justifies the computation of the quantile value using all the covariates, as explicited in Equation (4).

**Fixed-value threshold for the scoring function.** In the instantiation of GATE that we have used in the experiments, our main focus was to control the *number* of generated potential outcomes, and as a result we have decided to use a percentile-based definition of the scoring function $s(x, t)$ (where choosing $\alpha = 0.5$ guarantees that 50% of missing potential outcomes are generated, thus allowing to fix the proportion of generated outcomes across datasets).

However, in real-world applications a more optimal strategy might be to let $\alpha$ be a fixed variance threshold instead, the value of which can be guided by domain-knowledge or exploratory analysis of the data. This would more explicitly guardrail against the inclusion in the augmented dataset of particularly 'poor' generated outcomes. Then, we would define $\mathcal{X}_t = \{X_i \mid i \in [n], s(X_i, T_i) < \alpha_t\}$. We note that this in case, the proportion of generated outcomes depends on the properties of the generative model. In particular, if the model is particularly bad, no potential outcomes are generated and our method recovers the baseline performance.

# C  DETAILS OF THE THEORETICAL RESULTS

Let $f_t \in \mathcal{H}$ denote the hypothesis used to make the predictions for $Y(t)$, where $\mathcal{H} \subset \{h : \mathcal{X} \to \mathcal{Y}\}$ is a hypothesis class. Let $L : \mathcal{Y} \times \mathcal{Y} \to \mathbb{R}_+$ be a loss function (e.g. the squared loss function $L(y, y') = (y - y')^2$). We define the following quantities:

- Pointwise loss: $\ell_{f_t}(x) := \mathbb{E}_{Y(t)|X=x} \left[ L(Y(t), f_t(x)) \right]$,

- Marginal risk: $R(f_t) := \mathbb{E}_X \left[ \ell_{f_t}(X) \right] = \mathbb{E}_{Y(t),X} \left[ L(Y(t), f_t(X)) \right]$,

- Marginal risk for the augmented distribution: $\tilde{R}(f_t) := \mathbb{E}_{X'} \left[ \ell_{f_t}(X') \right] = \mathbb{E}_{Y'(t),X'} \left[ L(Y'(t), f_t(X')) \right]$,

- Factual risk: $R_t(f_t) := \mathbb{E}_{X|T=t} \left[ \ell_{f_t}(X) \right] = \mathbb{E}_{Y(t),X|T=t} \left[ L(Y(t), f_t(X)) \right]$,

- Counterfactual risk: $R_{1-t}(f_t) := \mathbb{E}_{X|T=1-t} \left[ \ell_{f_t}(X) \right] = \mathbb{E}_{Y(t),X|T=1-t} \left[ L(Y(t), f_t(X)) \right]$,

- Empirical risk on the augmented distribution: $\tilde{R}^{(\mathrm{emp})}(f_t) := \frac{1}{\tilde{n}_t} \sum_{i=1}^{\tilde{n}_t} L(Y_i'(t), f_t(X_i'))$.

In addition to these notations related to the risk, we define the class of functions $\mathcal{L} \subset \{x \to \mathbb{R}_+\}$ comprising functions $g : x \mapsto \mathbb{E}_{Y(t)|X=x} \left[ L(Y(t), f_t(x))|X = x \right]$ for all $f_t \in \mathcal{H}$. Furthermore, for any $x \in \mathcal{X}$, we define a class of functions $\mathcal{L}^x \subset \{\mathcal{Y} \to \mathbb{R}_+\}$ comprising the functions $l_{f_t}^x : y \mapsto L(y, f_t(x)) \in \mathcal{L}^x$ for all $f_t \in \mathcal{H}$. Finally, for a class of functions $\mathcal{S}$ and two distributions $P$ and $P'$, we write $\mathrm{IPM}_{\mathcal{S}}(P, P') = \sup_{f \in \mathcal{S}} |\mathbb{E}_{V \sim P}[f(V)] - \mathbb{E}_{W \sim P'}[f(W)]|$ for the Integral Probability Metric between $P$ and $P'$ defined for the class $\mathcal{S}$.

We first recall the statement of the generalization bound:

**Theorem C.1.** *(Full formulation of Theorem 3.1)  Assume access to an augmented dataset* $\{(X_i', Y_i'(t))\}_{i=1}^{\tilde{n}_t} \overset{i.i.d.}{\sim} P'(X', Y'(t))$ *and assume that* $0 < \mathbb{E}_{X',Y' \sim P'} \left[ L^2(Y', f_t(X')) \right] < +\infty$. *Then with probability at least* $1 - \delta$,

$$R(f_t) \leq \tilde{R}^{(\mathrm{emp})}(f_t) + (1 - \pi_t)\mathbb{E}_{X \sim Q} \left[ IPM_{\mathcal{L}^x} \left( P(Y(t) \mid X), P^{(gen)}(Y^{(gen)}(t) \mid X) \right) \right] \quad (5)$$

$$+ (1 - \pi_t)IPM_{\mathcal{L}} \left( Q, P_{1-t} \right) + V_{P'} \frac{\mathcal{C}_{\tilde{n}_t, \delta}^{\mathcal{H}}}{\tilde{n}_t^{3/8}}, \quad (6)$$

*where* $V_{P'} = \max\left(\sqrt{\mathbb{E}_{X',Y'(t)\sim P'}\left[L^2(Y'(t), f_t(X'))\right]}, \sqrt{\mathbb{E}_{X',Y'\sim \hat{P}'}\left[L^2(Y'(t), f_t(X'))\right]}\right)$, *with*

$\hat{P}'$ *denoting the empirical distribution for* $P'$, *and* $\mathcal{C}^{\mathcal{H}}_{\tilde{n}_t,\delta} = 2^{5/4}\left(\frac{d\log\frac{2e\tilde{n}_t}{d}+\log\frac{8}{\delta}}{\tilde{n}_t}\right)^{\frac{3}{8}}$, *with* $d$ *the pseudo-dimension of* $\{(x,y)\mapsto L(y, f_t(x)) \mid f_t \in \mathcal{H}\}$.

*Proof.* To prove Theorem C.1 for a given hypothesis $f_t$, our goal lies in obtaining a finite-sample generalisation bound of the marginal risk $R(f_t)$. We further note that:

$$R(f_t) = \pi_t R_t(f_t) + (1-\pi_t)R_{1-t}(f_t).$$

In this decomposition, $R_t(f_t)$ is the factual risk which is identifiable from the observational data under the ignorability assumption, and as such can be estimated using the empirical risk. However, $R_{1-t}(f_t)$ is not identifiable from the observational data. Thus, bounding the marginal risk is possible only after bounding the counterfactual risk, which requires accounting for the covariate shift and the variance in the outcomes, as we demonstrate below.

**Lemma C.2.** *Let* $f_t \in \mathcal{H}$. *The following inequality holds:*

$$R(f_t)-\tilde{R}(f_t) \leq (1-\pi_t)\left(\text{IPM}_{\mathcal{L}}\left(P_{1-t}, Q\right) + \mathbb{E}_{X\sim Q}\left[\text{IPM}_{\mathcal{L}^X}\left(P(Y(t)|X), P^{(gen)}(Y^{(\text{gen})}(t)|X)\right)\right]\right) \tag{7}$$

*Proof.* As stated earlier, $R(f_t) = \pi_t R_t(f_t) + (1-\pi_t)R_{1-t}(f_t)$. We obtain a similar decomposition for $\tilde{R}(f_t)$:

$$\tilde{R}(f_t) = \mathbb{E}_{X',Y'(t)}\left[L(Y'(t), f_t(X'))\right] \tag{8}$$

$$= \pi_t\mathbb{E}_{X',Y'(t)|A=t}\left[L(Y', f_t(X')\right] + (1-\pi_t)\mathbb{E}_{X',Y'(t)|A=1-t}\left[L(Y'(t), f_t(X')\right] \tag{9}$$

$$= \pi_t\mathbb{E}_{X,Y(t)|T=t}\left[L(Y, f_t(X)\right] + (1-\pi_t)\mathbb{E}_{X',Y'(t)|A=1-t}\left[L(Y'(t), f_t(X')\right] \tag{10}$$

$$= \pi_t R_t(f_t) + (1-\pi_t)\tilde{R}_{1-t}(f_t) \tag{11}$$

where line (10) follows by definition of $(A, X', Y'(t))$. Hence, $R(f_t) - \tilde{R}(f_t) = (1-\pi_t)\left(R_{1-t}(f_t) - \tilde{R}_{1-t}(f_t)\right)$.

We can then bound $R_{1-t}(f_t) - \tilde{R}_{1-t}(f_t)$ as follows:

$$R_{1-t}(f_t) - \tilde{R}_{1-t}(f_t) \tag{12}$$

$$= \mathbb{E}_{X,Y(t)|T=1-t}\left[L(Y(t), f_t(X))\right] - \mathbb{E}_{X',Y'|A=1-t}\left[L(Y'(t), f_t(X'))\right] \tag{13}$$

$$= \mathbb{E}_{X|T=1-t}\left[\mathbb{E}_{Y(t)|X}\left[L(Y(t), f_t(X))|X\right]\right] - \mathbb{E}_{X'|A=1-t}\left[\mathbb{E}_{Y'(t)|X',A=1-t}\left[L(Y'(t), f_t(X'))|X'\right]\right] \tag{14}$$

$$= \mathbb{E}_{X\sim P_{1-t}}\left[\mathbb{E}_{Y(t)|X}\left[L(Y(t), f_t(X))|X\right]\right] - \mathbb{E}_{X'\sim Q}\left[\mathbb{E}_{Y'(t)|X',A=1-t}\left[L(Y'(t), f_t(X') \mid X')\right]\right] \tag{15}$$

$$= \mathbb{E}_{X\sim P_{1-t}}\left[\mathbb{E}_{Y(t)|X}\left[L(Y(t), f_t(X))|X\right]\right] - \mathbb{E}_{X\sim Q}\left[\mathbb{E}_{Y(t)|X}\left[L(Y(t), f_t(X)) \mid X\right]\right] \tag{16}$$

$$+ \mathbb{E}_{X\sim Q}\left[\mathbb{E}_{Y(t)|X}\left[L(Y(t), f_t(X))|X\right]\right] - \mathbb{E}_{X\sim Q}\left[\mathbb{E}_{Y^{(\text{gen})}(t)|X}\left[L(Y^{(\text{gen})}(t), f_t(X)) \mid X\right]\right] \tag{17}$$

$$\leq \sup_{g\in\mathcal{L}}\left|\mathbb{E}_{X\sim P_{1-t}}\left[g(X)\right] - \mathbb{E}_{X\sim Q}\left[g(X)\right]\right| \tag{18}$$

$$+ \mathbb{E}_{X\sim Q}\left[\sup_{\ell^X\in\mathcal{L}^X}\left|\mathbb{E}_{Y(t)|X}\left[\ell^X(Y(t))\right] - \mathbb{E}_{Y^{(\text{gen})}(t)|X}\left[\ell^X(Y^{(\text{gen})}(t))\right]\right|\right] \tag{19}$$

$$= \text{IPM}_{\mathcal{L}}\left(P_{1-t}, Q\right) + \mathbb{E}_{X\sim Q}\left[\text{IPM}_{\mathcal{L}^X}\left(P(Y(t)|X), P^{(gen)}(Y^{(\text{gen})}(t)|X)\right)\right] \tag{20}$$

where in line (13) we used the fact that $\mathbb{E}_{X'\sim Q}\left[\mathbb{E}_{Y'(t)|X',A=1-t}\left[L(Y'(t), f_t(X') \mid X')\right]\right] = \mathbb{E}_{X\sim Q}\left[\mathbb{E}_{Y^{(\text{gen})}(t)|X}\left[L(Y^{(\text{gen})}(t), f_t(X)) \mid X\right]\right]$ (by definition of $Y'(t)$).

Multiplying the sum of the IPM terms by the factor $1-\pi_t$ then yields the result. □

Having bounded the difference in the marginal risk between the original $P$ and the augmented distribution $P'$, we now introduce the empirical risk to bound the marginal risk for $P'$.

**Lemma C.3.** *For any $f_t \in \mathcal{H}$, assume that $0 < \mathbb{E}_{X',Y' \sim P'}\left[L^2(Y', f_t(X'))\right] < +\infty$. Let $0 < \delta < 1$, and consider an augmented dataset $\{(X'_i, Y'_i(t))\}_{i=1}^{\tilde{n}_t} \overset{i.i.d.}{\sim} P'(X', Y'(t))$. The following bound then holds with probability at least $1 - \delta$:*

$$\tilde{R}(f_t) \leq \tilde{R}^{(\mathrm{emp})}(f_t) + V_{P'} \frac{\mathcal{C}^{\mathcal{H}}_{\tilde{n}_t,\delta}}{\tilde{n}_t^{3/8}} \tag{21}$$

*where $V_{P'} = \max\left(\sqrt{\mathbb{E}_{X',Y'(t) \sim P'}\left[L^2(Y'(t), f_t(X'))\right]}, \sqrt{\mathbb{E}_{X',Y' \sim \hat{P}'}\left[L^2(Y'(t), f_t(X'))\right]}\right)$, with $\hat{P}'$ denoting the empirical distribution for $P'$, and $\mathcal{C}^{\mathcal{H}}_{\tilde{n}_t,\delta} = 2^{5/4}\left(\frac{d \log \frac{2e\tilde{n}_t}{d} + \log \frac{8}{\delta}}{\tilde{n}_t}\right)^{\frac{3}{8}}$, with $d$ the pseudo-dimension of $\{(x, y) \mapsto L(y, f_t(x)) \mid f_t \in \mathcal{H}\}$.*

*Proof.* This result directly follows from Corollary 2 in the supplementary material of (Cortes et al., 2010). □

By summing the bounds involved in the Lemma C.2 and Lemma C.3, we then obtain Theorem C.1. □

# D EXPERIMENTAL DETAILS

## D.1 LICENSE FOR EXISTING ASSETS

The following existing assets were used to produce the experimental results:

- *Hillstrom* dataset (Hillstrom, 2008): available from https://blog.minethatdata.com/2008/03/minethatdata-e-mail-analytics-and-data.html
- *STAR Project* dataset (Achilles et al., 2008): CC0 1.0 License
- *Lalonde* dataset (LaLonde, 1986; Dehejia & Wahba, 1999; 2002): CC BY-NC 2.0 DEED License
- *RealCause* python library (Neal et al., 2020): MIT License
- *CATENets* python library (Curth & Van der Schaar, 2021a;b; Curth et al., 2021): BSD 3-Clause License
- *COCOA* code (Aloui et al., 2023)

## D.2 DATASET DETAILS

- **Lalonde** (LaLonde, 1986): The covariates comprise several demographic variables (e.g. age, degree, marital status). The treatment corresponds to attending a job training program. The outcome is the real earnings obtained in 1978. We generate the dataset using the trained models in (Neal et al., 2020).
- **STAR project** (Achilles et al., 2008): The individuals correspond to students, and we use the following covariates: *Gender, Race, Birth year, G3 Surban, G3 Free lunch, G3 Present, Aided class, G3 Teacher gender, G3 Teacher race, G3 Teacher high degree, G3 Teach years of experience, G3 Teacher training*, where G3 denotes Grade 3. The treatment corresponds to putting the student in a small class. In our analysis we have only included students who were assigned to the same treatment group through all grades K-3. The outcome is the SAT score of the student.
- **Hillstrom** (Hillstrom, 2008): The covariates correspond to different customers' attributes such as the months since last purchase or the zip code of the customer. The treatment corresponds to sending an email for men's merchandise. The outcome corresponds to whether or not the customer visited the website in the following two weeks.

Table 2: **Details on the datasets.**

| Dataset | Type of obs. dataset | # Samples (obs.) | Covariate dim. | Label |
|---------|----------------------|------------------|----------------|-------|
| Lalonde | Semi-synthetic | 7279 | 8 | Continuous |
| STAR project | Subsampled from RCT | 1429 | 12 | Continuous |
| Hillstrom | Subsampled from RCT | 9639 | 8 | Binary |

We provide an overview of the datasets' characteristics in Table 2.

**Dataset subsampling.** While the Lalonde dataset is semi-synthetic, the STAR and Hillstrom observational datasets used throughout the experiments in Section 6 are obtained by subsampling from their respective RCT data. We follow the same procedure as in (Gentzel et al., 2021), by defining a biasing function, with the desideratum that this biasing function should introduce a covariate shift between the treated and control groups. More precisely, given an original dataset $\{(X_i, T_i, Y_i) \mid i \in [n]\}$, we define an encoder $r$ such that $r(x)$ is the first PCA component score for $x$, obtained with the set of covariates $\{X_i \mid i \in [n]\}$. Given this encoder, we then compute $\gamma = \text{Median}(\{r(X_i) \mid i \in [n]\})$. This permits to construct the datasets $\mathcal{S}_0 = \{(X_j, T_j, Y_j) \mid r(X_j) < \gamma, j \in [n]\}$ and $\mathcal{S}_1 = \{(X_j, T_j, Y_j) \mid r(X_j) \geq \gamma, j \in [n]\}$. Intuitively, these two groups have a substantial difference in terms of covariates, as is captured by the encoder $r$. Finally, we obtain the observational dataset using the subsampling mechanism of (Gentzel et al., 2021) and keep the individuals in $\mathcal{S}_0$ with treatment equal to 0, and individuals in $\mathcal{S}_1$ with treatment equal to 1, i.e. $\mathcal{D}^{(\text{obs})} = \{(X_j, T_j, Y_j) \mid (X_j, T_j, Y_j) \in \mathcal{S}_0, T_j = 0\} \bigcup \{(X_j, T_j, Y_j) \mid (X_j, T_j, Y_j) \in \mathcal{S}_1, T_j = 1\}$.

In Section 6.5.1, we adjust the biasing intensity to modulate the covariate shift. To do so, we consider a probability $p \in [0, 1]$. We then define $\mathcal{D}^{(\text{obs})}(p) = \{(X_j, T_j, Y_j) \mid (X_j, T_j, Y_j) \in \mathcal{S}_0, B_j \sim \text{Ber}(1-p), T_j = B_j\} \bigcup \{(X_j, T_j, Y_j) \mid (X_j, T_j, Y_j) \in \mathcal{S}_1, B_j \sim \text{Ber}(p), T_j = B_j\}$.

Intuitively, higher values of $p$ yields a more pronounced covariate shift. We consider $p \in [0.5, 0.8, 1]$ in Section 6.5.1.

**Ground-truth CATE.** We fit two random forest models to half of the original and large STAR and Hillstrom datasets, which permits to estimate the two potential outcome surfaces for each of the datasets. This approach is not biased because these original datasets are RCTs. Equipped with the fitted potential surfaces, we then take their difference to define the ground-truth CATE values used for model evaluation. The other half of the datasets is then used to define an observational dataset (used to train the CATE learners) and a test set (used to evaluate the CATE learners).

### D.3    IMPLEMENTATION DETAILS FOR THE CATE LEARNERS

**Hardware.**    All the experiments were performed on a machine equipped with a 64-Core AMD Ryzen Threadripper and a NVIDIA RTX A4000. Fitting one CATE learner for one given dataset took in the worst case 3 minutes, and generating the augmented datasets with the LLM took a maximum of 17 minutes and 43 seconds per dataset and seed.

We now detail the hyperparameters used for the different CATE learners used in Section 6, which use neural networks backbones.

- **TNet**: Following (Curth & Van der Schaar, 2021b), each hypothesis function has 3 layers with 200 units. The output head consists of 2 additional layers with 100.
- **SNet**: We use 3 layers with 100 hidden units for the shared layers, 2 layers with 100 units for the output head of the hypothesis functions, and 2 layers with 100 units for the output of the propensity network.
- **XNet**: *First stage:* We use the $T$ strategy to estimate the nuisance parameters. We use 3 layers with 100 units for the representation, 2 layers with 100 units for the output head, *Second stage:* We use 2 layers with 100 units for the output, 3 layers with 200 units for the representation.
- **DRNet**: *First stage:* We use the $T$ strategy to estimate the nuisance parameters. We use 3 layers with 200 units for the representation, 2 layers with 100 units for the output head, *Second stage:* We use 2 layers with 100 units for the output, 3 layers with 200 units for the representation.

- **CFR-Wass**: We use 3 layers with 200 units for the representation layers and 3 layers with 100 units per hypothesis function. We use the Wasserstein-1 distance for the regularization, with the regularization coefficient $\alpha$ set to 3.
- **CFR-MMD**: We use 3 layers with 200 units for the representation layers and 3 layers with 100 units per hypothesis function. We use the MMD for the regularization, with the regularization coefficient $\alpha$ set to 3.
- **RNet**: We use 3 layers with 200 units for the representation, 2 layers with 100 units for the output head, for the two stages.
- **IPW**: *First stage:* We use the $T$ strategy to estimate the nuisance parameters. We use 3 layers with 200 units for the representation, 2 layers with 100 units for the output head, *Second stage:* We use 2 layers with 100 units for the output, 3 layers with 200 units for the representation.

The batch size is set to 500, the learning rate is set to 0.0001 with the Adam optimizer and we use early stopping with a validation split proportion equal to 0.3.

**LLM.** We use `GPT-4` as the LLM throughout our experiments, which we access using the API, version `2023-07-01-preview`. We use a temperature of 0.7 throughout our experiments.

## D.4 IMPLEMENTATION DETAILS FOR THE GENERATIVE MODELS

The following models perform augmentation by training $P_{0,x}^{(gen)}$ on $\mathcal{D}_0^{(obs)}$ and $P_{1,x}^{(gen)}$ on $\mathcal{D}_1^{(obs)}$ respectively. In particular:

- Mean imputation: $P_{t,x}^{(gen)} = \delta(\frac{1}{n_t} \sum_{i=1}^n Y_i \mathbb{1}(T_i = t))$, where $\{X_i, T_i, Y_i\} \in D_t^{(obs)}$

- Random Forest: $P_{t,x}^{(gen)} = \delta(f_t^{(RF)}(x))$, where $f_t^{(RF)}$ is a random forest model trained on $\mathcal{D}_t^{(obs)}$

- Nearest-neighbor: $P_{t,x}^{(gen)} = \delta(f_t^{(NN)}(x))$ where $f_t^{(NN)}$ is a nearest-neighbor predictor trained on $\mathcal{D}_t^{(obs)}$.

On the other hand, the GAN augmentation uses a single model trained on $\mathcal{D}^{(obs)}$. We refer to (Yoon et al., 2018) for the details of the method. We use the following parameters: { hidden dimension: 100, batch size: 256, iteration: 10000, $\alpha$ : 1, learning rate: 0.001 }

## D.5 METRICS

**Assessing covariate shift with the sliced Wasserstein distances.** In the experiments in Section 6.5.1 and Section 6.5.2, we quantify the covariate shift between the treated and control group using the sliced Wasserstein distance. It is a metric which can compare two high-dimensional distributions (Bonneel et al., 2015). To compute it, we perform random projections on vectors of the unit sphere. For two distributions $\mu_1$ and $\mu_2$, the sliced Wasserstein distance of order $p$ is defined as:

$$SW_p(\mu_1, \mu_2) := \int_{\mathbb{S}^{d-1}} W_p(P_u \# \mu_1, P_u \# \mu_2) du \tag{22}$$

where $\mathbb{S}^{d-1}$ denotes the unit sphere in dimension $d$, $P_u(x) = u \cdot x$ denotes the projection of the vector $x$ on $u$, $P_u \# \mu$ is the push-forward of $\mu$ by $P_u$, and $W_p$ is the Wasserstein distance of order $p$. In our experiments, we use a Monte-Carlo estimate by randomly sampling $n = 5000$ random vectors $\{u_i | i \in [n]\}$ in $\mathbb{S}^{d-1}$ and consider $p = 2$.

**Assessing the inaccuracy of the generated potential outcomes.** Let us consider an augmented dataset $\{(O_i, X_i', T_i', Y_i')\}_{i=1}^{\tilde{n}}$, where $T_i'$ denotes the observed (factual) treatment, $O_i = 0$ if $Y_i'$ is the observed potential outcome and $O_i = 1$ if $Y_i'$ was generated with $P_{t,x}^{(gen)}$. We assess in Section 6.5.2 the inaccuracy of the generated potential outcomes in the augmented dataset by computing:

$$\Delta = \frac{1}{\tilde{n}} \sum_{i=1}^{\tilde{n}} (Y_i' - \mathbb{E}[Y_i(1 - T_i') \mid X_i'])^2 \mathbb{1}(O_i = 1)$$

**PEHE.** Our results in Section 6 evaluate the performance of the models on $\mathcal{D}_{\text{test}}$ using the Precision in Estimation of Heterogeneous Effect (PEHE), defined as $\epsilon_{\text{PEHE}} = \frac{1}{n}\sum_{i=1}^{n}(\mathbb{E}\left[Y_i(1) - Y_i(0)|X = X_i\right] - (\hat{\mu}_1(X_i) - \hat{\mu}_0(X_i)))^2$. We report its square root $\sqrt{\epsilon_{\text{PEHE}}}$ (Hill, 2011).

# E   LLM PROMPTS

**Prompt design.** When instantiating GATE with LLMs, we consider a prompt structure which includes the following important elements:

- **Task context**: We include context about the task (CATE estimation). We also provide information about the covariates, the treatment, and the outcomes.
- **Statistics on the outcomes**: we provide the average outcomes in both the control and treatment group, as well as the range of the outcomes to help the LLM generate realistic outcomes.
- **In-context samples**: we serialize the observational data in their raw format. The covariates are provided as (feature name, feature value) tuples, followed by (treatment name, treatment value), and (outcome name, outcome value). The in-context samples are randomly shuffled in the prompt to avoid any generation artifacts stemming from the ordering of the samples. We use 100 in-context samples per prompt.

The prompt structure is summarized in Figure 6.

---

You are an expert in causal inference. Your goal is to produce counterfactuals from observational data. I will give you the covariates, the treatment and the outcome from the observational data. Leverage your knowledge about {Task context: general}. The covariates consist of {Task context: covariates description} The treatment indicator (binary) corresponds to { Task context: treatment description}. The outcome is { Task context: outcomes }. To help you, I am providing some statistics about the data. {Statistics treatment group} {Statistics control group} Your response should only contain the generated counterfactuals in the format ## outcome ##. {In-context examples}

---

Figure 6: Prompt structure.

**Prompt example.** We provide an example of the prompt used for the Lalonde dataset in Listing 1.

Listing 1: **Prompt example.** On `Lalonde` dataset.

```
You are an expert in causal inference. Your goal is to produce
    counterfactuals from observational data. I will give you the
    covariates, the treatment and the outcome from the
    observational data. Leverage your knowledge about job training
     and real earnings to produce counterfactuals. The covariates
    consist of a number of demographic variables: age, measured in
     years; education, measured in years; black, indicating race
    (1 if black, 0 otherwise);hispanic, indicating race (1 if
    Hispanic, 0 otherwise);married, indicating marital status (1
    if married, 0 otherwise); nodegree, indicating high school
    diploma (1 if no degree, 0 otherwise); re74, real earnings in
    1974; re75, real earnings in 1975. The treatment indicator (
    binary) corresponds to job training. The outcome is real
    earnings in the year 1978, denoted as re78. To help you, I am
    providing some statistics about the data. In the presence of
    the treatment (treat: 1), the average re78 (outcome) in the
    observational data is 4576.24, the min re78 is 0.0, the max
    re78 is 26354.16. In the absence of the treatment (treat: 0),
    the average re78 (outcome) in the observational data is
```

```
14868.48, the min re78 is 0.0, the max re78 is 28609.63. Your
   response should only contain the generated counterfactuals in
   the format ## outcome ##
Covariates: age: 26.0, education: 11.0, black: 0.0, hispanic: 0.0,
    married: 1.0, nodegree: 1.0, re74: 25862.32, re75: 16650.0
treat: 0
re78: ## 24058.61 ##
Covariates: age: 23.0, education: 7.0, black: 1.0, hispanic: 0.0,
    married: 1.0, nodegree: 1.0, re74: 18350.49, re75: 14967.1
treat: 0
re78: ## 8564.2 ##

...
Covariates: age: 30.0, education: 16.0, black: 0.0, hispanic: 0.0,
    married: 1.0, nodegree: 0.0, re74: 695.54, re75: 930.97
treat: 1
re78:
```

**No context prompt** We provide in Listing 2 the prompt used throughout Section 6.4, where the contextual information is removed.

Listing 2: **Prompt example without contextual information.** On `Lalonde` dataset.

```
You are an expert in causal inference. Your goal is to produce
   counterfactuals from observational data. I will give you the
   covariates, the treatment and the outcome from the
   observational data. To help you, I am providing some
   statistics about the data. In the presence of the treatment (
   treat: 1), the average re78 (outcome) in the observational
   data is 4576.24, the min re78 is 0.0, the max re78 is 26354.16.
    In the absence of the treatment (treat: 0), the average re78
   (outcome) in the observational data is 14868.48, the min re78
   is 0.0, the max re78 is 28609.63. Your response should only
   contain the generated counterfactuals in the format ## outcome
    ##
Covariates: Feature_0: 26.0, Feature_1: 11.0, Feature_2: 0.0,
   Feature_3: 0.0, Feature_4: 1.0, Feature_5: 1.0, Feature_6:
   25862.32, Feature_7: 16650.0
treat: 0
outcome: ## 24058.61 ##
Covariates: Feature_0: 23.0, Feature_1: 7.0, Feature_2: 1.0,
   Feature_3: 0.0, Feature_4: 1.0, Feature_5: 1.0, Feature_6:
   18350.49, Feature_7: 14967.1
treat: 0
outcome: ## 8564.2 ##

...
Covariates: Feature_0: 30.0, Feature_1: 16.0, Feature_2: 0.0,
   Feature_3: 0.0, Feature_4: 1.0, Feature_5: 0.0, Feature_6:
   695.54, Feature_7: 930.97
treat: 1
outcome:
```

**Dataset splitting.** Since the LLM context window limits the number of tokens which can be used in the prompt, we cannot feed all the available observational data into a single prompt. To bypass this issue, we randomly partition the observational dataset into different groups of in-context samples, each of these groups making one prompt. Each group is populated by $n_{\text{ICL}} = 100$ samples. Having split the observational data into different groups, we construct the prompts as follows. For each individual, we identify the group it belongs to, and construct a prompt where the individual appears at

the end of the prompt, with the rest of the group passed as in-context examples above it in a random order to avoid any ordering bias. The LLM then generates $m = 10$ outcomes for each individual and its associated constructed prompt.

**Memorization risks.** A natural question is whether or not the LLM is returning outcomes which have been memorized and seen during its pretraining stage. We note that this is very unlikely to be the case, since by definition, the LLM is used to output missing potential outcomes, which are not present in the observational datasets and hence not part of the pretraining corpora of the LLM. We also remark that the Lalonde dataset is semi-synthetic, meaning that it is also very unlikely that it has been memorized by the LLM.

# F ADDITIONAL RESULTS

## F.1 COMPARISON WITH ALOUI ET AL. (2023)

We compare GATE with COCOA (Aloui et al., 2023). As discussed in Appendix A, COCOA employs a local regression model which is trained on the observational data only. This limitation can make COCOA particularly susceptible to covariate shift scenarios or when operating in a small-sample regime, where the available data may not sufficiently capture the underlying distribution of outcomes.

We report the results in Table 3, comparing the LLM-instantiated GATE with COCOA, which shows that the LLM-instantiated GATE consistently outperforms COCOA across almost all of the datasets and meta-learners. The performance gap is particularly noticeable for the Lalonde dataset, where the control and treated groups are imbalanced, making the local regression model in COCOA significantly less useful than the prior-knowledge-empowered LLMs.

Table 3: **Comparison with COCOA (Aloui et al., 2023).** Performance comparison across the datasets for COCOA and GATE . Average $\sqrt{\epsilon_{\text{PEHE}}}$ and $1\text{std}$ is reported for 3 seeds ($\downarrow$ is better).

| Learner | Lalonde CPS1D | | STAR | | Hillstrom | |
|---|---|---|---|---|---|---|
| | COCOA | GATE | COCOA | GATE | COCOA | GATE |
| R-learner | 1.66±0.42 | 0.95±0.00 | 0.58±0.03 | 0.47±0.01 | 0.30±0.02 | 0.26±0.02 |
| IPW-learner | 1.12±0.05 | 0.95±0.01 | 0.59±0.07 | 0.47±0.01 | 0.34±0.11 | 0.25±0.00 |
| TARNet | 1.26±0.08 | 0.96±0.01 | 0.45±0.06 | 0.48±0.04 | 0.27±0.01 | 0.24±0.00 |
| DragonNet | 1.04±0.06 | 0.95±0.02 | 0.51±0.03 | 0.48±0.04 | 0.27±0.01 | 0.24±0.01 |
| CFR-MMD | 1.01±0.01 | 0.95±0.00 | 0.58±0.15 | 0.44±0.00 | 0.24±0.00 | 0.24±0.00 |
| BART | 1.32±0.01 | 1.35±0.00 | 0.62±0.07 | 0.56±0.02 | 0.26±0.01 | 0.25±0.01 |
| T-learner | 1.35±0.06 | 0.96±0.01 | 0.66±0.08 | 0.50±0.03 | 0.28±0.03 | 0.24±0.01 |
| S-learner | 1.04±0.14 | 0.95±0.01 | 0.88±0.13 | 0.56±0.02 | 0.28±0.02 | 0.25±0.01 |
| X-learner | 1.38±0.15 | 0.95±0.01 | 0.73±0.04 | 0.49±0.02 | 0.27±0.01 | 0.24±0.01 |
| DR-learner | 1.35±0.05 | 0.95±0.01 | 0.62 ± 0.2 | 0.48±0.02 | 0.31±0.02 | 0.25±0.01 |
| CFR-Wass. | 0.98±0.04 | 0.95±0.02 | 0.55±0.15 | 0.41±0.01 | 0.24 ± 0.0 | 0.24 ± 0.0 |

## F.2 COMPARISON ON THE IHDP DATASET

We evaluate the benefits of GATE instantiated with an LLM for the IHDP dataset (Shalit et al., 2017). We note that the outcomes for this dataset are synthetic. Therefore, the objective of this experiment is to assess the in-context learning abilities of the LLM, and the importance of covariate shift reduction via data augmentation. We report the results in Table 4, showing that GATE improves the performance of almost all the CATE learners.

## F.3 SENSITIVITY WITH RESPECT TO $\alpha$

We complement the results shown in Section 6.5.2, with Figure 7 and Figure 8, which present the impact of varying the quantile $\alpha$ used to define the admissible set $\mathcal{X}_t$. We note that the results of the trade-off experiment presented here (Figure 8) and in the main text (Figure 5) were obtained using the DR-learner. For both the STAR Project and Hillstrom datasets, we see that incorporating the generated outcomes helps improve the PEHE. However, unlike for the Lalonde dataset, there is

Table 4: **Comparison on IHDP.** Performance comparison for the IHDP dataset, between No augmentation and GATE . Average $\sqrt{\epsilon_{\text{PEHE}}}$ and 1std is reported for 3 seeds (↓ is better).

| Learner | IHDP | |
|---|---|---|
| | No aug. | GATE |
| S-learner | $0.71 \pm 0.10$ | $0.54 \pm 0.03$ |
| T-learner | $0.70 \pm 0.13$ | $0.40 \pm 0.06$ |
| X-learner | $0.68 \pm 0.10$ | $0.33 \pm 0.04$ |
| R-learner | $0.68 \pm 0.04$ | $0.37 \pm 0.01$ |
| IPW-learner | $0.85 \pm 0.04$ | $0.38 \pm 0.04$ |
| DR-learner | $0.61 \pm 0.06$ | $0.37 \pm 0.04$ |
| TARNet | $0.47 \pm 0.03$ | $0.31 \pm 0.04$ |
| DragonNet | $0.41 \pm 0.02$ | $0.31 \pm 0.04$ |
| CFR-MMD | $0.29 \pm 0.01$ | $0.27 \pm 0.01$ |
| CFR-Wass. | $0.28 \pm 0.01$ | $0.29 \pm 0.05$ |
| BART | $0.56 \pm 0.00$ | $0.59 \pm 0.01$ |

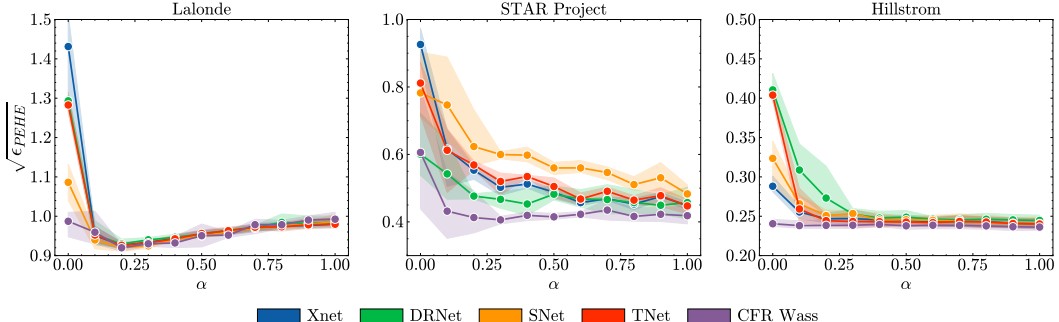

Figure 7: Sensitivity with respect to $\alpha$

no clear cutoff value for $\alpha$ after which the PEHE starts increasing. This observation can be made more intuitive by examining Figure 8. Indeed, we notice that the covariate shift reduction obtained by increasing $\alpha$ is less pronounced than in the case of the Lalonde dataset, while the noise introduced with the generated outcomes increases at a similar rate. This explains why setting higher values of $\alpha$ is not harmful: the effect of the reduction in covariate shift balances the increased inaccuracy in the generated potential outcomes.

### F.4 COMPARISON WITH THE NON-PARAMETRIC BASELINES

We provide additional results for the experiment conducted in Section 6.3, where we do not perform selection for the baselines (meaning that we set $\mathcal{X}_t = \mathcal{X}$ for all the baselines, except for the LLM). We report the results in Figure 9, which confirms the performance gains obtained by using LLMs as the generative model in GATE. We note that the results of the comparison experiment presented

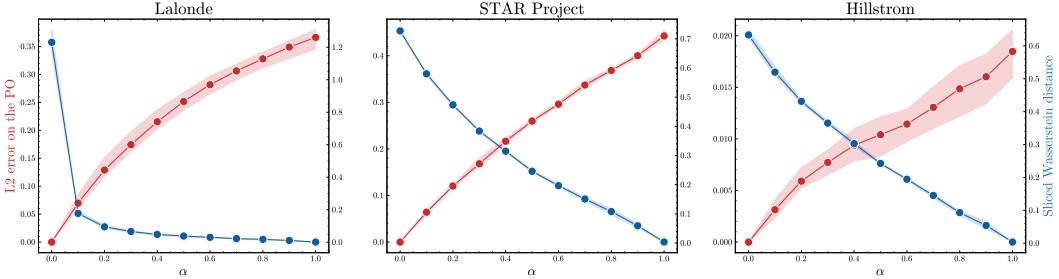

Figure 8: Tradeoff between covariance shift and potential outcome generation inaccuracy

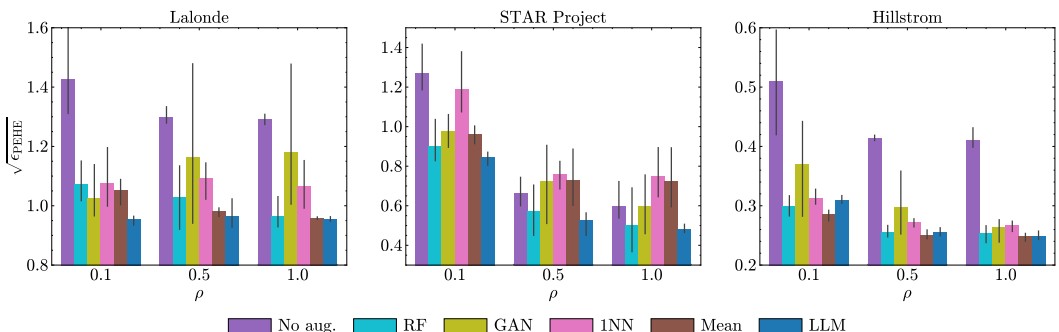

Figure 9: Comparison of the LLM with other non-parametric baselines (no selection)

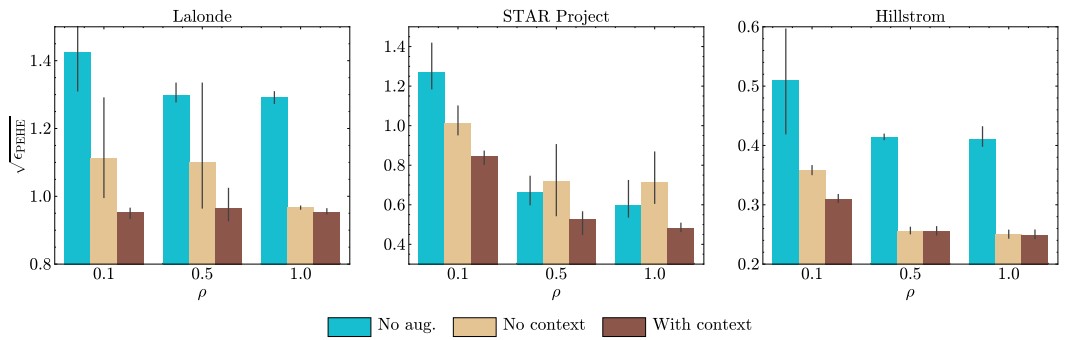

Figure 10: Comparison of using DR-learner fitted on the data augmented with `GATE`, using the LLM prompted with and without context. The error bars mark 1std computed over 3 seeds.

here (Figure 9) and in the main text (Figure 3) were obtained using the DR-learner. We notice that the performance gap on the Hillstrom dataset is negligible. This aligns with our observation that the treatment effect is very small for this dataset. Indeed, the Average Treatment Effect, defined as $\mathbb{E}\left[Y(1) - Y(0)\right]$ is equal to 0.08. Furthermore, the variability in the outcome is negligible, with $\mathrm{Var}(Y(1)) = 0.04$ and $\mathrm{Var}(Y(0)) = 0.02$, explaining why the mean imputation baseline performs competitively with respect to the LLM. In contrast, the ATE for the STAR dataset is equal to 0.15, and $\mathrm{Var}(Y(1)) = 1.05$ and $\mathrm{Var}(Y(0)) = 0.93$ (computed on the normalized outcomes), where the larger variability explains the performance gap between the LLM and the mean baseline.

### F.5 IMPORTANCE OF CONTEXTUAL INFORMATION

Following the same experimental setup as in Section 6.4, we assess the importance of the contextual information to improve the potential outcome generation for the Lalonde and Hillstrom datasets. We report the results in Figure 10. We note that the results of the context experiment presented here (Figure 2) and in the main text (Figure 10) were obtained using the DR-learner. For the Lalonde dataset, we notice that the gains obtained using contextual information are especially noticeable in the small-sample regime (i.e. $\rho = 0.1$), echoing the observations made for the STAR Project dataset. The performance gap narrows down with an increasing $\rho$, as the increased sample size in factual data makes the CATE learner more robust with respect to the inaccuracy of the generated potential outcomes. On the other hand, the performance gap on the Hillstrom dataset is negligible. This aligns with our observation that the treatment effect is very small for this dataset.

# G    ADDITIONAL REBUTTAL RESULTS

## G.1    STATISTICAL TESTS OF IMPROVEMENTS

**Experiment setting.** In order to assess the statistical significance of the results in Table 1, we conduct two-sample t-tests on the $\sqrt{\epsilon_{PEHE}}$ obtained with and without GATE (instantiated with LLMs).

**Results.** We report the p-values in Table 5, showing that the performance gains obtained with GATE are statistically significant at the 0.05 level across the majority of CATE learners and datasets.

Table 5: **Statistical significance of GATE's performance gains.** We report the p-values of the two-sample t-tests, where bolded entries represent statistical significance at the 0.05 level.

| Learner | Lalonde CPS1D | STAR | Hillstrom |
|---|---|---|---|
| S-learner | $\mathbf{3.0 \times 10^{-2}}$ | $\mathbf{2.2 \times 10^{-2}}$ | $\mathbf{1.3 \times 10^{-2}}$ |
| T-learner | $\mathbf{4.0 \times 10^{-5}}$ | $\mathbf{4.0 \times 10^{-3}}$ | $\mathbf{1.3 \times 10^{-5}}$ |
| X-learner | $\mathbf{1.3 \times 10^{-3}}$ | $\mathbf{1.5 \times 10^{-4}}$ | $\mathbf{4.1 \times 10^{-3}}$ |
| R-learner | $\mathbf{1.1 \times 10^{-2}}$ | $\mathbf{1.5 \times 10^{-7}}$ | $\mathbf{4.9 \times 10^{-4}}$ |
| IPW-learner | $\mathbf{1.0 \times 10^{-4}}$ | $\mathbf{1.9 \times 10^{-5}}$ | $\mathbf{7.4 \times 10^{-9}}$ |
| DR-learner | $\mathbf{6.8 \times 10^{-6}}$ | $1.3 \times 10^{-1}$ | $\mathbf{1.4 \times 10^{-4}}$ |
| CFR-Wass. | $3.7 \times 10^{-1}$ | $\mathbf{3.4 \times 10^{-6}}$ | $3.8 \times 10^{-1}$ |
| CFR-MMD. | $\mathbf{4.9 \times 10^{-6}}$ | $1.6 \times 10^{-1}$ | $6.0 \times 10^{-1}$ |
| TARNet | $\mathbf{1.1 \times 10^{-5}}$ | $2.8 \times 10^{-1}$ | $\mathbf{3.2 \times 10^{-16}}$ |
| DragonNet | $1.7 \times 10^{-1}$ | $\mathbf{7.3 \times 10^{-6}}$ | $\mathbf{1.8 \times 10^{-7}}$ |
| BART | $\mathbf{1.3 \times 10^{-6}}$ | $6.6 \times 10^{-2}$ | $3.0 \times 10^{-1}$ |

## G.2    ALTERNATIVE SELECTION OF IN-CONTEXT SAMPLES

**Experimental setting.** We consider an instantiation GATE with LLM where the in-context samples used in the prompts are $k$ nearest-neighbours of the samples considered for augmentation. More specifically, given a sample $(x, t)$, we define $\mathcal{S}_{x,t} = \mathrm{NN}_k(X, \mathcal{D}_{1-t}^{(obs)})$ as the set of in-context samples for $(x, t)$. We set $k = 50$ and use a DR-learner for downstream CATE estimation.

**Results.** As presented in Table 6, our results demonstrate that random sampling of in-context samples from $\mathcal{D}^{(obs)}$ (encompassing both control and treated groups) consistently yields superior performance compared to the nearest neighbor baseline. This is intuitive given the covariate shift between the two groups, which inherently limits the utility of nearest-neighbor information drawn from the opposing treatment group. Random sampling, by contrast, enables the incorporation of individuals from both groups – a particularly advantageous approach when prior knowledge exists regarding the relationship between $Y^1$ and $Y^0$ (e.g. difference in expectation).

Table 6: **Comparison of in-context samples' selection.** Results reported for 3 seeds.

| IC sampling | Lalonde CPS1D | STAR | Hillstrom |
|---|---|---|---|
| | $\rho = 0.1$ | | |
| Nearest neighbor | $1.09 \pm 0.12$ | $0.99 \pm 0.08$ | $0.39 \pm 0.06$ |
| Random sampling | $0.95 \pm 0.02$ | $0.85 \pm 0.04$ | $0.31 \pm 0.01$ |
| | $\rho = 0.5$ | | |
| Nearest neighbor | $1.10 \pm 0.04$ | $0.62 \pm 0.09$ | $0.28 \pm 0.02$ |
| Random sampling | $0.97 \pm 0.05$ | $0.53 \pm 0.07$ | $0.26 \pm 0.01$ |
| | $\rho = 1$ | | |
| Nearest neighbor | $1.09 \pm 0.10$ | $0.58 \pm 0.02$ | $0.26 \pm 0.01$ |
| Random sampling | $0.95 \pm 0.01$ | $0.48 \pm 0.02$ | $0.25 \pm 0.01$ |

### G.3 COMPARISON AGAINST OTHER SELECTORS

**Experimental setting.** We compare the variance-based selector used in our LLM instantiation of GATE with two additional selectors: (1) a selector which selects the samples uniformly at random in the observational dataset (*Random*) and (2) a propensity-based selector (*Propensity*), which defines the score function as $s(x, t) = P(T = t | X = x)$, intuitively favouring samples exhibiting characteristics similar to those from the opposite treatment group. For all the selectors, we set $\alpha = 0.5$, and use a DR-Learner for downstream CATE estimation.

**Results.** We report the results in Table 7, showing that the variance-based selector achieves optimal performance most consistently out of the considered selection criteria, with performance gains especially noticeable in the small-sample regime ($\rho = 0.1$).

Table 7: **Comparison against other selectors.** Results reported for 3 seeds.

| Selector | Lalonde CPS1D | STAR | Hillstrom |
|---|---|---|---|
| | $\rho = 0.1$ | | |
| *Random* | $0.95 \pm 0.02$ | $0.90 \pm 0.21$ | $0.35 \pm 0.03$ |
| *Propensity* | $0.95 \pm 0.01$ | $0.87 \pm 0.14$ | $0.39 \pm 0.02$ |
| *Variance* | $0.95 \pm 0.02$ | $0.85 \pm 0.04$ | $0.31 \pm 0.01$ |
| | $\rho = 0.5$ | | |
| *Random* | $1.00 \pm 0.06$ | $0.54 \pm 0.02$ | $0.25 \pm 0.01$ |
| *Propensity* | $1.00 \pm 0.04$ | $0.50 \pm 0.07$ | $0.35 \pm 0.00$ |
| *Variance* | $0.97 \pm 0.05$ | $0.53 \pm 0.07$ | $0.26 \pm 0.01$ |
| | $\rho = 1$ | | |
| *Random* | $0.98 \pm 0.01$ | $0.49 \pm 0.03$ | $0.25 \pm 0.01$ |
| *Propensity* | $1.02 \pm 0.06$ | $0.47 \pm 0.10$ | $0.33 \pm 0.03$ |
| *Variance* | $0.95 \pm 0.01$ | $0.48 \pm 0.02$ | $0.25 \pm 0.01$ |

