# OpenReview forum: "Enhancing Treatment Effect Estimation with Generation-Driven Data Augmentation"
_ICLR.cc/2025/Conference — Submitted to ICLR 2025_

### Official Review · Reviewer_xENs · 2024-10-30

**Soundness:** 2
**Presentation:** 2
**Contribution:** 2
**Rating:** 5
**Confidence:** 4

**Summary:**

The paper addresses the estimation of conditional average treatment effects (CATE) from observational data using a framework called GATE. It employs generative models to augment datasets with synthetic potential outcomes, focusing on mitigating covariate shift. The key contribution is demonstrating that targeted data augmentation can improve CATE estimation, especially in low-data scenarios.

**Strengths:**

1. The experiments conducted are extensive and well-supported.

2. The main content of the paper is clearly written and easy to follow.

**Weaknesses:**

1. First, I have some questions about the originality of this paper; data augmentation for causal data is not a novel concept, and I do not see any significant differences from existing works in the authors' theoretical discussions and method design.

2. The paper primarily focuses on qualitative analysis regarding covariate shift but lacks quantitative assessments to substantiate the claims that the augmented data effectively mitigates covariate shift. This absence of rigorous empirical validation undermines the overall effectiveness of the proposed method.

3. The depiction and explanation of Figure 1 are quite ambiguous and do not effectively convey the key points of the method. I recommend making some revisions for clarity.

**Questions:**

1.	In Figure 1, it appears that the symbol \( k \) is not utilized. Is this an oversight or is there a specific reason for its absence? I kindly request clarification from the authors, as this is crucial for understanding your methodology.

2.  Can you provide a more intuitive example to explain how your causal data augmentation method differs from general data augmentation and other causal enhancement methods?

---

> ### Author Response · Authors · 2024-11-22
> **Response to Reviewer xENs (1/2)**
>
> *We appreciate the reviewer’s detailed and thoughtful feedback.*
>
> ---
>
> ## Clarifying the Contribution of this Work
>
> Let us clarify the key novelties of our work in the following points.
>
> **Problem setting.** Our work addresses a particularly challenging yet overlooked scenario in CATE estimation: the *small-sample regime* under *high covariate shift* between treatment groups. As demonstrated in  `Section 6`, conventional approaches which rely on model specifications (e.g. inverse-propensity weighting, representation learning) typically struggle in this setting.
>
> **General augmentation framework.** To address this problem, we introduce $\texttt{GATE}$, a general and model-agnostic framework to improve *any* CATE learner via data augmentation. $\texttt{GATE}$ is flexible, since it allows to leverage *diverse* generative models. Our theoretical analysis in `Section 3.1` reveals the benefits of the $\texttt{GATE}$ framework, namely an increase of the effective sample size and the mitigation of covariate shift, which can offset potential inaccuracies of the generative model.
>
> **Difference with traditional data augmentation.**  Furthermore, the $\texttt{GATE}$ framework contrasts traditional data augmentation in that it does not require the generation of covariates, a complex task which could introduce additional covariate shift. Rather, we rely on generating the missing potential outcomes for already existing covariates. This approach directly reduces the covariate shift and as such, can provide performance benefits which can be strong enough to counterbalance the bias potentially introduced by an imperfect generative model. This distinguishes $\texttt{GATE}$ from traditional data augmentation, methods where inaccurate generative models can easily degrade performance (Manousakas & Aydöre, 2023).
>
> **External knowledge integration.** Motivated by the inherent limitations of small observational datasets, we instantiate $\texttt{GATE}$ with LLMs to illustrate the advantages of generative models which incorporate external sources of knowledge. This key novelty in the context of CATE estimation represents a significant departure from existing imputation approaches, such as Aloui et al. (2023), which are ultimately contrained by the observational data (e.g. using local regression models). We further introduce a new selection mechanism based on the LLM's uncertainty to define the admissible set $\mathcal{X}_{t}$ used for augmentation.
>
> **Empirical performance.** Our comprehensive set of experiments shows that this LLM-based instantiation of $\texttt{GATE}$ has the following benefits: 1) it consistenly improves the PEHE across $11$ CATE learners and $3$ datasets  2) it outperforms other instantiations of $\texttt{GATE}$ which use generative models trained solely on $D^{(obs)}$, with performance gains especially noticeable in the small-sample regime ($\rho=0.1$). In `Section 6.4` we provide additional insights showing the importance of eliciting the prior knowledge of LLMs to achieve optimal performance gains. Further, in `Section 6.5` we provide a detailed analysis verifying the theoretical insights about data augmentation for CATE estimation presented in `Section 3`.
>
> **We acknowledge that these key contributions were not stated clearly enough in our original manuscript.** Hence, we have revised it to clarify its key novelty, with specific changes detailed in our global response.
>
> ---
>
> ## Quantitative assessment verifying the covariate shift reduction
>
> We would like to point out that the following experiments, *present in our original manuscript*, illustrate that the augmented data effectively mitigates covariate shift.
>
> **Quantifying the reduction in covariate shift.** The experiment described in `Section 6.5.2` demonstrates how the covariate shift between the control and treated groups decreases  with increasing values of the parameter $\alpha$, as evidenced in `Figure 5 Right`. We quantified the distributional distance between the control and treated groups by computing the Sliced Wasserstein (SW) distance between the covariates in the control and treated groups. We provide more details about this metric in `Appendix D.4`.
>
> **GATE is particularly helpful in high covariate shift settings.** Having shown that $\texttt{GATE}$ permits to effectively mitigate covariate shift, we demonstrated in `Section 6.5.1` and `Figure 5 Left` that its performance gains were the most pronounced when the covariate shift between the treated and control groups in the observational dataset is important.
>
> Therefore, these two experiments confirm the theoretical insights derived from our generalization bound (`Theorem 3.1`): augmenting the observational data with $\texttt{GATE}$ allows to reduce the covariate shift, hence leading to improved CATE estimation.

---

> ### Author Response · Authors · 2024-11-22
> **Response to Reviewer xENs (2/2)**
>
> ## Other comments
>
> **Clarifying $Y_{i,k}^{(gen)}$:** We account for the potential stochastic nature of generative models (e.g. LLMs) by sampling $K$ outcomes $\{ Y_{i,k}^{(gen)} | k=1, ..., K \}$ from the distribution $P_{t, x_i}^{(gen)}$ for a given covariate $x_i$.
> We then aggregate this set into a single outcome, by computing the average $\frac{1}{K} \sum_{i=1}^{K} Y_{i,k}^{(gen)}$
> , which defines $\bar{Y}_{i}^{(gen)}$.
>
> We recognize that this was not clear in our original manuscript, which is why we have added an extended clarification at `l.264`.
>
> **Improvements to Figure 1.** Thank you for pointing out the lack of clarity of `Figure 1`. We have made the changes to the notation used in the figure, as well as in the figure caption, to ensure that Figure 1 allows to comprehensively understand our work. Please let us know if there are any further suggestions that we can incorporate.
>
>
> ---
> *We hope the reviewer’s concerns are addressed and they will consider updating their score. We welcome further discussions.*
>
> ### References
> Manousakas, Dionysis, and Sergül Aydöre. "On the usefulness of synthetic tabular data generation." arXiv preprint arXiv:2306.15636 (2023).

---

> > ### Comment · Reviewer_xENs · 2024-11-25
> >
> > Thank you for your detailed response. After careful consideration, I find the mentioned issues persist and will keep my score unchanged.

---

### Official Review · Reviewer_w9Rs · 2024-11-03

**Soundness:** 3
**Presentation:** 2
**Contribution:** 2
**Rating:** 3
**Confidence:** 4

**Summary:**

The paper presents a generative modeling based approach to impute missing potential outcomes in a subset of the covariate space. It presents one general theoretical result supporting the general methodology of augmentation as well as empirical studies.

**Strengths:**

The paper presents a new generative modeling approach to impute a reliable subset of the missing potential outcomes.

**Weaknesses:**

The main weakness lies in the incremental nature of the contribution. While using LLMs to generate a subset of missing potential outcomes is novel, the core concept of identifying a subset of individuals within the feature space for imputation has been previously explored, as in Aloui 2023 (which is mentioned by the authors) and Nagalapatti 2024 (https://arxiv.org/pdf/2401.15447). More extensive studies on the effects of various generative models and a comparison of their performance are needed to make the work more comprehensive. Moreover, The code was not added in the supplementary materials, adding even a notebook with few experiments, would have have been helpful to assess the reproducibility of the results.

**Questions:**

1. How do the authors tune the parameter $\alpha$, in real world experiments only the factual data is availabe and estimating the epehe to validate is not possible, how do the authors intend to deal with this?

2. If the selected generative model (or the LLM) used has an inherent bias due its training can it bias the imputation as well?

3. Can the authors explain the variance problem? And how is it related to the imputation error? I can have deterministic imputation (by imputing 0 to every missing potential outcomes) hence it has a zero variance but what matters more is the imputation error? Shouldn't the tradeoff by between imputation error and covariate shift between treatment groups?

4. How sensitive are the imputation for changing the prompts in both the in context and no context prompts?

---

> ### Author Response · Authors · 2024-11-22
> **Response to Reviewer w9Rs (Part 1/3)**
>
> *We appreciate the reviewer’s detailed and thoughtful feedback.*
>
> ---
>
> ## Clarifying the Contribution of this Work
>
> We respectfully disagree that the contribution of our work is "incremental", and that its core contributions have been "previously explored". Let us clarify the key novelties in the following points.
>
> **Problem setting.** Our work addresses a particularly challenging yet overlooked scenario in CATE estimation: the *small-sample regime* under *high covariate shift* between treatment groups. As demonstrated in  `Section 6`, conventional approaches which rely on model specifications (e.g. inverse-propensity weighting, representation learning) typically struggle in this setting.
>
> **General augmentation framework.** To address this problem, we introduce $\texttt{GATE}$, a general and model-agnostic framework to improve *any* CATE learner via data augmentation.  $\texttt{GATE}$ is flexible, since it allows to leverage *diverse* generative models. Our theoretical analysis in `Section 3.1` reveals the benefits of the $\texttt{GATE}$ framework, namely an increase of the effective sample size and the mitigation of covariate shift, which can offset potential inaccuracies of the generative model.
>
> **External knowledge integration.** Motivated by the inherent limitations of small observational datasets, we instantiate $\texttt{GATE}$ with LLMs to illustrate the advantages of generative models which incorporate external sources of knowledge. This key novelty in the context of CATE estimation represents a significant departure from existing imputation approaches, such as Aloui et al. (2023), which are ultimately contrained by the observational data (e.g. using local regression models). We further introduce a new selection mechanism based on the LLM's uncertainty to define the admissible set $\mathcal{X}_{t}$ used for augmentation.
>
> **Empirical performance.** Our comprehensive set of experiments shows that this LLM-based instantiation of $\texttt{GATE}$ has the following benefits: 1) it consistenly improves the PEHE across $11$ CATE learners and $3$ datasets  2) it outperforms other instantiations of $\texttt{GATE}$ which use generative models trained solely on $D^{(obs)}$, with performance gains especially noticeable in the small-sample regime ($\rho=0.1$). In `Section 6.4` we provide additional insights showing the importance of eliciting the prior knowledge of LLMs to achieve optimal performance gains. Further, in `Section 6.5` we provide a detailed analysis verifying the theoretical insights about data augmentation for CATE estimation presented in `Section 3`.
>
> **We acknowledge that these key contributions were not stated clearly enough in our original manuscript.** Hence, we have revised it to clarify its key novelty, with specific changes detailed in our global response.
>
> **Additional experimental contributions.** In addition to clarifying the contributions, we have also followed the suggestion of the reviewer to provide a "more extensive study" of generative models. In light of this, we focused on our instantiation of $\texttt{GATE}$ with LLMs, and analyzed the sensitivity of $\texttt{GATE}$ to the prompt construction with two additional experiments:
> * *Number of in-context samples*: in `Section 6.4, Figure 4`, we vary the number of in-context samples provided in the prompt, showing that $\texttt{GATE}$ instantiated with LLMs benefits from the use of a wide context to elicit the LLM's in-context learning abilities.
> * *Selection of in-context samples*: in `Appendix G.2`, we show that random sampling of in-context samples (among both the treated and control group) outperforms a nearest-neighbor baseline which only considers the opposite treatment group. This demontrates  better in-context learning capabilities of LLM when leveraging information from *both* treated and control groups under high covariate shift.

---

> ### Author Response · Authors · 2024-11-22
> **Response to Reviewer w9Rs (Part 2/3)**
>
> ## Clarifying the Variance-Based Definition of the Admissible Set
>
> **Variance and imputation error.** Let us stress that our proposed variance-based selector is specifically meant for the instantiation of $\texttt{GATE}$ with stochastic models trained on external datasets, such as LLMs, and is based on their unique characteristics. Indeed, in the context of LLMs, it has been shown that uncertainty measures such as variance can be used to discriminate between factually correct and incorrect responses (Huang et al., 2023; Manukal et al., 2023), as well as predict the quality of a response (Lin et al., 2023), thus validating the use of variance as the selection criterion. In case of deterministic models trained on $\mathcal{D}^{(obs)}$, rather than relying on variance (which then indeed is equal to zero), we propose to rely on other metrics, such as the propensity score (as we explain in `l.293` in our manuscript).
>
> **Comparison with other selectors.** We performed an additional experiment, detailed in `Appendix G.3`, to compare the variance-based selector with (1) a random selector and (2) a propensity-based selector, where we define the score function as $s(x,t) = P(T = t|X=x)$.
> We report the results obtained with the LLM instantiation of $\texttt{GATE}$ in `Table 7`, fixing $\alpha = 0.5$. These results show that the variance-based selector achieves optimal performance most consistently out of the considered selection criteria, with performance gains especially noticeable in the small-sample regime ($\rho = 0.1$).
>
>
> **Tuning the hyperparameter $\alpha$.** The tuning of the hyperparameter $\alpha$ is linked to the broader problem of model selection in CATE estimation. Due to the fundamental problem of causal inference, true counterfactuals are unobservable in real-world experiments, preventing straightforward model selection through validation set performances. Therefore, model selection procedures for causal inference models remain an active area of research (Lan & Syrgkanis, 2024; Curth et al., 2023; Mahajan et al., 2022; Saito & Yasui, 2020; Schuler et al., 2018), and advances in this field will naturally benefit $\texttt{GATE}$.
>
> With these challenges in mind, we propose three complementary strategies which allow to guide the selection of the value of $\alpha$ within the $\texttt{GATE}$ framework:
> 1. As we explain above, relying on the **fixed-threshold definition of the admissible set** (rather than the percentile-based definition of the threshold) can allow to guide the selection of $\alpha$ using domain knowledge.
> 2. We further propose to guide the selection of $\alpha$ by **measuring the covariate shift** between the sets $\tilde{\mathcal{D}}_0^{(obs)}$ and $\tilde{\mathcal{D}}_1^{(obs)}$, using for example the sliced Wasserstein distance (an example of such an analysis can be found in our `Figure 5 Right`). Then, we propose to choose the _minimal_ value of $\alpha$ which allows to achieve significant reduction in a covariate shift (which might correspond to the 'elbow' in the graph). While such an 'elbow' might not always exist, this criterion provides additional guidelines in certain circumstances.
> 3. Finally, the choice of $\alpha$ can be further guided by **standard methods for CATE model selection**, particularly those based on comparing the _downstream CATE models_ using a pseudo-outcome surrogate criteria evaluated on a held-out validation set (see Curth et al. (2023) for an overview). In particular, in view of strong covariate shift and small sample regime, we propose to rely on criteria which _do not_ rely on estimating the propensity score, as these might lead to high variance in such cases.
>
> Nevertheless, while finding the *optimal* value of $\alpha$ is non-trivial, our experiments showed the following: (1) Fixing $\alpha = 0.5$ consistently led to improved dowstream PEHE across the $3$ datasets and the $11$ CATE learners (cf. `Table 1`) (2) The strong effect of the reduction in covariate shift obtained with data augmentation (shown in `Figure 5 Right`) reduces the sensitivity with respect to $\alpha$. Indeed,`Figure 5 Middle` highlights that _any_ value $\alpha > 0$ leads to performance gains compared to $\alpha = 0$.

---

> ### Author Response · Authors · 2024-11-22
> **Response to Reviewer w9Rs (Part 3/3)**
>
> ## Bias with data augmentation
>
> We thank the reviewer for mentioning the potential bias of generative models. Let us explain how the practitioner using the LLM-instantiated $\texttt{GATE}$ has various tools at their disposal to address this point:
>
> **Auditing the generation of outcomes**: Contrasting other generative models, the use of LLMs with $\texttt{GATE}$ permits to make the generation process more transparent. Beyond producing numerical values, LLMs can also elicit explicit causal reasoning chains underpinning outcome generation when prompted accordingly. This capability enables human-in-the-loop applications of $\texttt{GATE}$, where domain experts can evaluate the generated data by examining these reasoning traces against their domain knowledge. As such, we view the use prompting techniques for explicit reasoning, such as chain-of-through (Wei et al., 2022) or tree-of-thought (Yao et al., 2024), as a promising direction for future work.
>
> **Fairness analysis:** Since $\texttt{GATE}$ is an augmentation method, practitioners can directly assess potential bias in the generated potential outcomes: (1) by doing a standard fairness analysis and computing different fairness metrics (Van Breugel et al., 2021; Dwork et al., 2012; Hardt et al., 2016; Mehrabi et al., 2021) on the augmented data. (2) If some bias is detected, the practitioner could use off-the-shelf debiasing methods (Calmon et al., 2017; Feldman et al., 2015).
>
> **Action Taken:** We have added a discussion about the potential bias of the LLMs into our `Appendix B.2`.
>
> ---
> ### Code release
> We provide an anonoymized version of our code at the following link: https://anonymous.4open.science/r/GATE-8849
>
> ---
> *We hope the reviewer’s concerns are addressed and they will consider updating their score. We welcome further discussions.*

---

> ### Author Response · Authors · 2024-11-22
> **Response to Reviewer w9Rs (References)**
>
> ### References
>
> Van Breugel, Boris, et al. "Decaf: Generating fair synthetic data using causally-aware generative networks." Advances in Neural Information Processing Systems 34 (2021): 22221-22233.
>
> Calmon, F. et al. (2017). Optimized pre-processing for discrimination prevention. Advances in neural information processing systems, 30.
>
> Curth, Alicia, and Mihaela Van Der Schaar. "In search of insights, not magic bullets: Towards demystification of the model selection dilemma in heterogeneous treatment effect estimation." International Conference on Machine Learning. PMLR, 2023.
>
> Dwork, C. et al. (2012). Fairness Through Awareness. In Proceedings of the 3rd Innovations in Theoretical Computer Science Conference (Cambridge, Massachusetts) (ITCS '12). ACM, New York, NY, USA, 214–226.
>
> Feldman, M. et al. (2015). Certifying and removing disparate impact. In proceedings of the 21th ACM SIGKDD international conference on knowledge discovery and data mining (pp. 259-268).
>
> Hardt, M. et al. (2016). Equality of opportunity in supervised learning. In Advances in neural information processing systems. 3315–3323.
>
> Huang, Yuheng, et al. "Look before you leap: An exploratory study of uncertainty measurement for large language models." arXiv preprint arXiv:2307.10236 (2023).
>
> Lan, Hui, and Vasilis Syrgkanis. "Causal q-aggregation for cate model selection." International Conference on Artificial Intelligence and Statistics. PMLR, 2024.
>
> Lin, Zhen, Shubhendu Trivedi, and Jimeng Sun. "Generating with confidence: Uncertainty quantification for black-box large language models." arXiv preprint arXiv:2305.19187 (2023).
>
> Mahajan, Divyat, et al. "Empirical Analysis of Model Selection for Heterogeneous Causal Effect Estimation." arXiv preprint arXiv:2211.01939 (2022).
>
> Manakul, Potsawee, Adian Liusie, and Mark JF Gales. "Selfcheckgpt: Zero-resource black-box hallucination detection for generative large language models." arXiv preprint arXiv:2303.08896 (2023).
>
> Mehrabi, N. et al. (2021). A survey on bias and fairness in machine learning. ACM computing surveys (CSUR), 54(6), 1-35.
>
> Saito, Yuta, and Shota Yasui. "Counterfactual cross-validation: Stable model selection procedure for causal inference models." International Conference on Machine Learning. PMLR, 2020.
>
> Schuler, Alejandro, et al. "A comparison of methods for model selection when estimating individual treatment effects." arXiv preprint arXiv:1804.05146 (2018).
>
> Wei, Jason, et al. "Chain-of-thought prompting elicits reasoning in large language models." Advances in neural information processing systems 35 (2022): 24824-24837.
>
> Yao, Shunyu, et al. "Tree of thoughts: Deliberate problem solving with large language models." Advances in Neural Information Processing Systems 36 (2024).

---

> > ### Comment · Reviewer_w9Rs · 2024-11-25
> >
> > I thank the authors for their thorough comments and I think the answered most of my concerns. However, I still believe that the work needs major additions to be complete, as the main contribution is emprical, a thorough comparison with other LLMs and generative models is needed in my opinion. I will hence keep my score.

---

### Official Review · Reviewer_tfVD · 2024-11-03

**Soundness:** 3
**Presentation:** 3
**Contribution:** 2
**Rating:** 5
**Confidence:** 5

**Summary:**

This work proposes a data augmentation method GATE to improve CATE estimation. In particular, GATE first uses generative models to generate missing potential outcomes, and select only a subset of the generated outcomes to augment the observational dataset. These augmented missing potential outcomes reduces the covariate shift problem as long as the bias of the generative model can be controlled. The authors consider multiple generative models including the LLMs and demonstrate the efficacy of GATE on multiple benchmarks.

**Strengths:**

- CATE estimation is an important problem with numerous applications.
- The proposed method is sound, both intuitively and theoretically.
- This paper is well-written.
- Figure 1 is very helpful for understanding the proposed method.
- The experiments are comprehensive.
- The consideration of *external data source* generative model (i.e., LLM) is very promising

**Weaknesses:**

## Originality
This work is based on the principle that **augmenting the observational dataset through potential outcome imputation on a selected subset of individuals** can reduce covariate shift, and if the bias of the imputation is small, then the augmented observational dataset can benefit any downstream CATE model. This principal is **identical** to the an one-year-old work [1] submitted to ICLR 2024 (with an arXiv version [2]), which also proposes a data augmentation method for CATE estimation.

Although this work uses different phrases, e.g., *"augmenting the observational dataset with carefully selected missing potential outcomes sampled from a generative model"* in this work vs. *"it performs imputation for the counterfactual outcomes of these selected individuals"* in [1], the principal is **exactly the same**. This can be further evidenced by the highlighted sentences in two works:
- In line 62 in this manuscript, *"benefits which can be strong enough to counterbalance the bias potentially introduced by an imperfect generative model"*.
- At the beginning of page of [1], *"the positive impact of disparity reduction will outweigh the negative impact of imputation error"*.

## Almost Identical Theoretical Result
The whole theory section (Section 3) of this work is **uncannily similar** to the theory section (Section 4.2) of [1]. In particular, both work motivates the data augmentation methods through generalization bounds of  CATE models. Both of the generalization bounds from these two works (Theorem 3.1 in this work and Prop 3 in [1]) include three terms:
- (i) a factual loss term (this work calls it the empirical risk and includes an extra finite-sample term established by another work);
- (ii) a statistical distance term that measures the distance between two measures on the covariates $X$ (this work uses IPM while [1] uses Total Variation);
- (iii) a statistical distance term involving the true potential outcome and the imputed potential outcome (this work uses IPM distance while [1] uses the L_2 distance between the true and estimated potential outcome functions).

While it is fine for theoretical results to appear similar, given the similarity of their motivations and principles, and that they are addressing the same problem (both this work and [1] propose data augmentation methods), I think **a detailed discussion and comparison with Prop 3 in [1]** is definitely needed. Also note that this is **the only theoretical result** in this work.

## Almost Identical Insights from the Theoretical Result

Given the similar theoretical result, it is not surprising that the insights of this work is similar to that of [1]. For example, just to name a few,
- *"tuning Q via the admissible set Xt allows to navigate the trade-off involved in data
augmentation'* in this work versus
- *"this theorem provides a rigorous illustration of the trade-off
between the statistical disparity across treatment groups and the imputation error"* in [1],

and
- *"excluding from Xt regions where the generative model performs poorly can
further enhance performance"* in this work versus
- *"It underscores that by simultaneously minimizing disparity and imputation error, we can enhance the performance of CATE estimation model"* in [1]

## Technical Weakness: Selection of the Missing Potential Outcome
The term 2 in Theorem 3.1 is interpreted as the noise of the generator, which consists of **both bias and variance**. The author proposes to select the generated outcomes with the variance, **completely ignoring the bias**.

[1] "Counterfactual Data Augmentation with Contrastive Learning" (https://openreview.net/forum?id=7mR83Q12cJ).

[2] Aloui, Ahmed, et al. "Counterfactual Data Augmentation with Contrastive Learning." arXiv preprint arXiv:2311.03630 (2023).

**Questions:**

- What percentage of the generated potential outcomes are eventually used to augment the observational dataset? If my understanding is correct, the percentage should be exactly $\alpha$.
- Are the variances of LLMs really highly correlated with their bias on all the benchmarks?
- How are the hyper parameters selected? For example, how do you choose the $\alpha$ in practice?

**Details Of Ethics Concerns:**

The **motivation, insights, and theoretical contribution of this work is extremely similar to an one-year-old work**  [1] which was submitted to ICLR 2024. Please see the *Weakness* section for a detailed discussion. This missing acknowledgement of [1] is considerably uncanny given that this works in fact cites [2], which is a historical arXiv version of [1] and includes the same motivation, insights, and theoretical contribution already.

From my perspective, **this work GATE should be considered as a generalization of [1,2]** where GATE treats the missing potential outcome imputation model as a general component that can be instantiated with various generative models including LLMs, and investigates methods to identity the reliable generated outcomes from LLMs.

Overall, I think this work needs heavy editing during rebuttal to acknowledge [1] and expand sections about LLMs, which I believe is the main contribution of this work. **If this was done appropriately**, I am more than willing to lean towards acceptance based on the **additional contribution** of this work on top of [1,2].

[1] "Counterfactual Data Augmentation with Contrastive Learning" (https://openreview.net/forum?id=7mR83Q12cJ).

[2] Aloui, Ahmed, et al. "Counterfactual Data Augmentation with Contrastive Learning." arXiv preprint arXiv:2311.03630 (2023).

---

> ### Author Response · Authors · 2024-11-22
> **Response to Reviewer tfVD (Part 1/4)**
>
> *We appreciate the reviewer’s detailed and thoughtful feedback.*
>
> ---
>
>
> ## Originality
>
> Following your suggestions, we have revised our manuscript to emphasize the paper's main contributions, including the use of external sources of data to enhance CATE estimation, which is possible through the LLM-based instantiation of $\texttt{GATE}$. We outlined in details the steps taken to achieve that in the general comment, posted above. All the introduced changes can be found in the revised version of the paper, which we have also already uploaded to OpenReview (changes are marked in teal). We thank you for the suggestion to introduce these changes, as we believe that they clarify the novelty and contributions of $\texttt{GATE}$.
>
> ---
>
> ## Theoretical results
> On top of the LLM-refocus, in our paper we have added an _extended_ discussion comparing our theoretical results to Aloui et al. (2023), with **additional references** to this work (and ideas presented within) incorporated into the `Introduction` (`l.064`), `Section 3` (`l.207`), `Section 4.1` (`l.239`), and `Section 4.2` (`l.294`).
>
> However, we still believe that the result we presented in `Theorem 3.1` is a fundamental part of our paper, theoretically validating our approach, and further expanding the previous results presented by Aloui et al. (2023), as we outline below. Further, we note that our `Theorem 3.1`, while similar, was derived using entirely different approach, relying on different metrics (IPM vs Total Variation and L2 distance) and different results from previous works (this work relies on Johansson et al. (2022), while Aloui et al. rely on Ben-David et al. (2010)).
>
> We believe that the result we presented (`Theorem 3.1`) is indeed a generalisation and extension of the result presented in Aloui et al. (2023), with the following key differences:
> 1. The formulation of the generalisation bound presented in this work accommodates an **arbitrary loss function** $L$.
> 2. `Theorem 3.1` presented in this work explicitly accounts for the potential **stochasticity of generative models** (such as LLMs), by employing an IPM term to measure the distributional distance between the ground-truth outcome and the generator's distribution.
> 3. Our bound characterizes the **finite-sample regime**, showing how the benefits of data augmentation correlate with sample size.
>
> ---
>
> ## Comparison of Insights from the Theoretical Results
> Further, we believe that the insights drawn from the theoretical results also differ between the two works. In particular:
> 1. We focus on the **small sample regime**, emphasising that efficient data augmentation can be particularly beneficial in the case when the observational data is small (we further verify this insight experimentally in `Section 6.3`).
> 2. We recognise that data augmentation becomes particularly important in problems with a **strong covariate shift**, which arises for example when the treatment is guided by expert judgment (we further verify this insight experimentally in `Section 6.5`).
> 3. We emphasise that the selection can be guided not by properties of the dataset, but rather by the **properties of the generative model**, which is particularly relevant when using generative models _not_ trained on $\mathcal{D}^{(obs)}$.
>
> ---
> ## Additional experimental results
> To further empirically analyse the LLM-based instantiation of $\texttt{GATE}$, we have also empirically investigated the sensitivity of the LLM-instantiated $\texttt{GATE}$ to the prompt construction by conducting two additional experiments:
> * *Number of in-context samples*: in `Section 6.4, Figure 4`, we vary the number of in-context samples provided in the prompt, showing that $\texttt{GATE}$ instantiated with LLMs benefits from the use of a wide context to elicit the LLM's in-context learning abilities.
> * *Selection of in-context samples*: in `Appendix G.2`, we show that random sampling of in-context samples (among both the treated and control group) outperforms a nearest-neighbor baseline which only considers the opposite treatment group. This demontrates  better in-context learning capabilities of LLM when leveraging information from *both* treated and control groups under high covariate shift.
>
> ---
>
> ## Selection of the Missing Potential Outcomes
>
> **The difficulty of evaluating the bias of the generative model.** As you have pointed out, our `Theorem 3.1` indicates that to maximise the performance gains obtained by using $\texttt{GATE}$, we should aim to exclude from the admissible set those regions of the covariate space, where $P_{t, x}^{(gen)}$ significantly deviates from $P(Y(t)|X=x)$. However, this requires estimating the distance between these two _distributions_, which is highly non-trivial, as it would require us to obtain multiple samples from $P(Y(t)|X=x)$ for each value of $x$. This is particularly infeasible under strong covariate shift and in the small-sample regime: both cases which call for the use of $\texttt{GATE}$.

---

> ### Author Response · Authors · 2024-11-22
> **Response to Reviewer tfVD (Part 2/4)**
>
> **Motivation for using a variance-based selection criterion.** To overcome this problem, we have proposed to use a _proxy measure_ to quantify the fidelity of $P_{t, x}^{(gen)}$ and efficiently identify regions which should be excluded from the admissible set $\mathcal{X}_t$. In particular, we rely on the uncertainty in the generated potential outcomes (measured with variance). While, as you point out, such a measure might not _always_ be the optimal choice for all generative models, in the context of LLMs it has been shown that uncertainty measures such as variance can be used to discriminate between factually correct and incorrect responses (Huang et al., 2023; Manakul et al., 2023), as well as predict the quality of a response (Lin et al., 2023), further validating our choice.
>
> **Correlation between variance and imputation errors.** Following your question, we have computed the Spearman correlation coefficient between the variance of the generated potential outcomes and the mean squared error on the predicted counterfactual. The results are presented in the table below.
>
> | Selector | Spearman Correlation | p-value |
> | -------- | -------- | -------- |
> | Lalonde CPS1D| $-0.02$ | $6 \times 10^{-4}$ |
> | STAR | $0.14$ | $7 \times 10^{-21}$ |
> | Hillstrom | $0.19$ | $4\times 10^{-27}$ |
>
>
> We note that the correlation between the variance and the mean squared error on the generated counterfactual outcomes are significantly larger than zero for the Hillstrom and STAR datasets, both of which are datasets with real-world outcomes obtained by subsampling a RCT dataset. In contrast, the missing potential outcomes in the Lalonde CPS1D are obtained using a black-box generative model (see `Appendix D.2` for an extended discussion of the construction of the datasets).
>
> **Cautionary remark: lack of direct link between the imputation error and PEHE.** In aiding the interpretation of the above results, we would like to clarify why relying on imputation error in the generated counterfactuals (for example computing an imputation MSE) to _evaluate_ the selection mechanism might _not_ be an optimal strategy. Indeed, the imputation MSE is not necesarily equivalent to the distributional distance between $P(Y(t) | X=x)$ and $P(Y^{(gen)}(t) | X=x)$, averaged over the different values of $X=x$, which is ultimately the quantity impacting downstream performance. In other words, two given generative models can achieve similar imputation MSE but lead to drastically different downstream PEHE.
>
> We empirically validate this insight with an additional experiment where we compare the performance of two generative models:
> - Model A:  adds Gaussian noise with standard deviation $\sigma = \sqrt{5000}$ to the true counterfactual outcomes.
> - Model B: adds the constant $\sigma$ to the true counterfactual outcomes.
>
> By construction, with a reasonable number of samples, the imputation MSE of Model A will be close to $\sigma^2$, which is the imputation MSE of Model B.
>
> We use the STAR dataset and for each model, we augment the observational dataset with its corresponding generated outcomes. We use a DR-Learner and repeat the experiment across $3$ different seeds.
>
>  The following table summarizes the results:
>
> | Model | Imputation MSE | $\sqrt{\epsilon_{PEHE}}$  |
> | -------- | -------- | -------- |
> | Model A | $4.9 \times 10^{3}\pm 0.07 \times 10^{3}$ | $0.21 \pm 0.01$ |
> | Model B | $5 \times 10^{3} \pm 0$ |  $0.57 \pm 0.02$  |
>
> Despite having similar MSE, the two models lead to very different downstream $\sqrt{\epsilon_{PEHE}}$, showing that imputation error should be interpreted carefully before being used for example to evaluate a selection mechanism.
>
> ---
> **Comparison to other selection mechanisms.** To evaluate the performance of the variance selection criterion more robustly, we have compared it against _random selection_ and _propensity score selction_ (where in the latter case we define $s(x,t) = P(T = t|X=x)$). We have included the details of this experiment in `Appendix G.3`, as well as presented them in the table below.
>
> #### Proportion $\rho =0.1$
> | Selector | Lalonde CPS1D | STAR | Hillstrom |
> | -------- | -------- | -------- |-------- |
> | Random | $0.95 \pm 0.02$ | $0.90 \pm 0.21$ | $0.35 \pm 0.03$ |
> | Propensity | $0.95 \pm 0.01$ | $0.87 \pm 0.14$ | $0.39 \pm 0.02$ |
> | Variance | $0.95 \pm 0.02$ | $0.85 \pm 0.04$ | $0.31 \pm 0.01$ |

---

> ### Author Response · Authors · 2024-11-22
> **Response to Reviewer tfVD (Part 3/4)**
>
> #### Proportion $\rho =0.5$
> | Selector | Lalonde CPS1D | STAR | Hillstrom |
> | -------- | -------- | -------- |-------- |
> | Random | $1.00 \pm 0.06$ | $0.54 \pm 0.02$ | $0.25 \pm 0.01$ |
> | Propensity | $1.00 \pm 0.04$ | $0.50 \pm 0.07$ | $0.35 \pm 0.00$ |
> | Variance | $0.97 \pm 0.05$ | $0.53 \pm 0.07$ | $0.26 \pm 0.01$ |
>
> #### Proportion $\rho =1.0$
> | Selector | Lalonde CPS1D | STAR | Hillstrom |
> | -------- | -------- | -------- |-------- |
> | Random | $0.98 \pm 0.01$ | $0.49 \pm 0.03$ | $0.25 \pm 0.01$ |
> | Propensity | $1.02 \pm 0.06$ | $0.47 \pm 0.10$ | $0.33 \pm 0.03$ |
> | Variance | $0.95 \pm 0.01$ | $0.48 \pm 0.02$ | $0.25 \pm 0.01$ |
>
>
> The results demonstrate that using the variance-based selection can lead to performance improvements, particularly in the small sample regime ($\rho = 0.1$). Further, the variance-based selection provides optimal performance most consistently out of the three methods considered.
>
> **Action Taken:** Following your comment, we have now added the results of the experiments presented above to `Appendix G.3`, with a reference to this section in `l. 303` in the revised manuscript. Further, we have included a discussion of the variance-based selection in `Appendix B.3`.
>
> ---
>
> ## Other questions
>
> **Size of the augmented dataset.** In the instantiation of $\texttt{GATE}$ that we have used in the experiments, our main focus was to control the _number_ of generated potential outcomes, and as a result we have decided to use a percentile-based definition of the scoring function $s(x, t)$ (where choosing $\alpha = 0.5$ guarantees that 50\% of missing potential outcomes are generated, thus allowing to fix the proportion of generated outcomes across datasets).
>
> However, in real-world applications a more optimal strategy might be to let $\alpha$ be a fixed variance threshold instead, the value of which can be guided by domain-knowledge or exploratory analysis of the data. This would more explicitly guardrail against the inclusion in the augmented dataset of particularly 'poor' generated outcomes. Then, we would define $\mathcal{X}_{t} = \\{X_i \mid i \in [n], s(X_i, T_i) < \alpha_t\\}$. We note that this in case, the proportion of generated outcomes depends on the properties of the generative model. In particular, if the model is particularly bad, no potential outcomes are generated and our method recovers the baseline performance.
>
> **Action Taken:** We have added a footnote about the fixed-value threshold for the scoring function in `l.322` with extended discussion in `Appendix B.3`.
>
> ---
>
> **Tuning the hyperparameter $\alpha$.** The tuning of the hyperparameter $\alpha$ is linked to the broader problem of model selection in CATE estimation. Due to the fundamental problem of causal inference, true counterfactuals are unobservable in real-world experiments, preventing straightforward model selection through validation set performances. Therefore, model selection procedures for causal inference models remain an active area of research (Lan & Syrgkanis, 2024; Curth et al., 2023; Mahajan et al., 2022; Saito & Yasui, 2020; Schuler et al., 2018), and advances in this field will naturally benefit $\texttt{GATE}$.
>
> With these challenges in mind, we propose three complementary strategies which allow to guide the selection of the value of $\alpha$ within the $\texttt{GATE}$ framework:
> 1. As we explain above, relying on the **fixed-threshold definition of the admissible set** (rather than the percentile-based definition of the threshold) can allow to guide the selection of $\alpha$ using domain knowledge.
> 2. We further propose to guide the selection of $\alpha$ by **measuring the covariate shift** between the sets $\tilde{\mathcal{D}}_0^{(obs)}$ and $\tilde{\mathcal{D}}_1^{(obs)}$, using for example the sliced Wasserstein distance (an example of such an analysis can be found in our Figure 5, right). Then, we propose to choose the _minimal_ value of $\alpha$ which allows to achieve significant reduction in a covariate shift (which might correspond to the 'elbow' in the graph). While such an 'elbow' might not always exist, this criterion provides additional guidelines in certain circumstances.
> 3. Finally, the choice of $\alpha$ can be further guided by **standard methods for CATE model selection**, particularly those based on comparing the _downstream CATE models_ using a pseudo-outcome surrogate criteria evaluated on a held-out validation set (see Curth et al. (2023) for an overview). In particular, in view of strong covariate shift and small sample regime, we propose to rely on criteria which _do not_ rely on estimating the propensity score, as these might lead to high variance in such cases.

---

> ### Author Response · Authors · 2024-11-22
> **Response to Reviewer tfVD (Part 4/4)**
>
> Nevertheless, while finding the *optimal* value of $\alpha$ is non-trivial, our experiments showed the following: (1) Fixing $\alpha = 0.5$ consistently led to improved dowstream PEHE across the $3$ datasets and the $11$ CATE learners (cf. `Table 1`) (2) The strong effect of the reduction in covariate shift obtained with data augmentation (shown in `Figure 5 Right`) reduces the sensitivity with respect to $\alpha$ in high covariate shift settings. Indeed,`Figure 5 Middle` highlights that _any_ value $\alpha > 0$ leads to performance gains compared to $\alpha = 0$.
>
> **Action Taken:** We have added an extended discussion about hyperparameter tuning to `Appendix B.2`.
>
> **Reference to [1] and [2].** As the reviewer noted, our original manuscript cited [2], with references in our related work sections (`Section 5`, `Appendix A`) and an experimental comparison (in `Appendix F.2` of the original manuscript). We chose to cite [2] instead of [1] since (a) the publication date of [2] is more recent than the publication date of [1], (b) we did not want to cause confusion by citing the same work twice.
>
> ---
>
> *We hope the reviewer’s concerns are addressed and they will consider updating their score. We welcome further discussions.*
>
> ---
>
> ### References
>
> Aloui, Ahmed, et al. "Counterfactual Data Augmentation with Contrastive Learning." arXiv preprint arXiv:2311.03630 (2023).
>
> Curth, Alicia, and Mihaela Van Der Schaar. "In search of insights, not magic bullets: Towards demystification of the model selection dilemma in heterogeneous treatment effect estimation." International Conference on Machine Learning. PMLR, 2023.
>
> Huang, Yuheng, et al. "Look before you leap: An exploratory study of uncertainty measurement for large language models." arXiv preprint arXiv:2307.10236 (2023).
>
> Johansson, Fredrik D., et al. "Generalization bounds and representation learning for estimation of potential outcomes and causal effects." Journal of Machine Learning Research 23.166 (2022): 1-50.
>
> Lan, Hui, and Vasilis Syrgkanis. "Causal q-aggregation for cate model selection." International Conference on Artificial Intelligence and Statistics. PMLR, 2024.
>
> Lin, Zhen, Shubhendu Trivedi, and Jimeng Sun. "Generating with confidence: Uncertainty quantification for black-box large language models." arXiv preprint arXiv:2305.19187 (2023).
>
> Mahajan, Divyat, et al. "Empirical Analysis of Model Selection for Heterogeneous Causal Effect Estimation." arXiv preprint arXiv:2211.01939 (2022).
>
> Manakul, Potsawee, Adian Liusie, and Mark JF Gales. "Selfcheckgpt: Zero-resource black-box hallucination detection for generative large language models." arXiv preprint arXiv:2303.08896 (2023).
>
> Saito, Yuta, and Shota Yasui. "Counterfactual cross-validation: Stable model selection procedure for causal inference models." International Conference on Machine Learning. PMLR, 2020.
>
> Schuler, Alejandro, et al. "A comparison of methods for model selection when estimating individual treatment effects." arXiv preprint arXiv:1804.05146 (2018).

---

> > ### Comment · Reviewer_tfVD · 2024-11-26
> >
> > Thank you for your response. It has addressed most of my concerns. I have updated my score.
> >
> > However, I agree with the remaining concerns from the fellow reviewers. To further increase score, they should be addressed first

---

### Official Review · Reviewer_iCno · 2024-11-04

**Soundness:** 3
**Presentation:** 2
**Contribution:** 2
**Rating:** 5
**Confidence:** 3

**Summary:**

The paper introduces a novel data augmentation approach for Rubin causal inference, specifically designed to address covariate shifts in Conditional Average Treatment Effect (CATE) estimation. The authors evaluate the effectiveness of augmenting only a subset of the covariate space, demonstrating that their method generally improves performance. This enhancement is attributed to the flexibility inherent in the GATE framework.

**Strengths:**

- The paper provides a straightforward approach for data augmentation in CATE estimation using LLM.
- It provides a comprehensive set of experiments to showcase the improvement over non-augmented approaches.

**Weaknesses:**

- The theoretical analysis lacks a clear connection to the design choices made for the GATE framework.
- Lacks the results for more recent CATE learners, such as TARNet [1], BART [2].
- No clear explanation on how to set up and train the generative models or utilize the LLM for GATE framework.
- The datasets used in these experiments are mostly synthetic datasets.
- No clear improvement when using augmentation in most cases.

[1] Shalit, Uri, Fredrik D. Johansson, and David Sontag. "Estimating individual treatment effect: generalization bounds and algorithms." International conference on machine learning. PMLR, 2017.
[2] Souto, Hugo Gobato, and Francisco Louzada Neto. "K-Fold Causal BART for CATE Estimation." arXiv preprint arXiv:2409.05665 (2024).

**Questions:**

- In section 4.1, how is the generative model set up when training on $D^{(obs)}_t$? Particularly, what type of generative model is used in this case? How does that affect the performance?
- Does the model only train to impute the mean outcome in that mean-imputing generative model?
- The scoring function directly influences the size of the selected set $X_t$. How exactly does this scoring function work when the generative model performs badly?

---

> ### Author Response · Authors · 2024-11-22
> **Response to Reviewer  iCno (Part 1/3)**
>
> *We appreciate the reviewer’s detailed and thoughtful feedback.*
>
> ---
> ## Results for more recent CATE learners
>
> We agree with the reviewer that evaluating the benefits of $\texttt{GATE}$ across various CATE learners is important. However, let us clarify that in addition to the results presented in `Table 1` (with 7 CATE learners), our original manuscript included results for TARNet (Shalit et al., 2017), DragonNet (Shi et al., 2019), CFR-MMD (Shalit et al., 2017) and BART (Athey & Imbens, 2016), presented in `Appendix F.1.` of our original manuscript. We realized that there was a wrong reference to this Appendix in the main text and apologize for this error.
>
> For clarity, we present these results below. We note the following: (1) $\texttt{GATE}$ instantiated with LLMs leads to significant performance improvements across most datasets and models, (2) it allows to decrease the performance gap between the different CATE models, making it a great data-preprocessing step which could potentially minimise the effect of model selection.
>
> #### Supplementary results for Table 1:
> | Learner | Lalonde CPS1D | STAR |Hillstrom|
> |-----|-----|-----|-----|
> |TARNet|❌1.27 +- 0.11 | ❌0.47 +- 0.07| ❌0.38 +- 0.01|
> ||✔️ 1.03 +- 0.06| ✔️ 0.43 +- 0.05| ✔️ 0.24 +- 0.00 |
> |DragonNet|❌1.11 +- 0.17 |❌ 0.88 +- 0.22 |❌0.40 +- 0.06 |
> ||✔️ 1.02 +- 0.06|✔️0.44 +-0.05|✔️ 0.24 +- 0.00|
> |CFR-MMD|❌1.00 +- 0.06 | ❌0.62 +- 0.09 | ❌0.24 +- 0.00 |
> ||✔️ 1.02 +- 0.06 | ✔️ 0.41 +- 0.03| ✔️ 0.24 +- 0.00 |
> |BART | ❌ 1.36 +- 0.05 | ❌0.69 +- 0.06 | ❌0.26+- 0.02|
> | | ✔️ 1. 37 +- 0.03| ✔️ 0.54 +- 0.03| ✔️ 0.25 +- 0.01|
>
> **Action Taken:** Following your comment, we have moved the results on the additional learners into the main text, adding them to `Table 1`.
>
>
> ---
>
> ## Details of how to utilise the LLM for the $\texttt{GATE}$ framework
>
> **Details on LLM instantiation.** Following your comment, we have dedicated `Section 4.1` in the revised version of the manuscript to detailing what strategies we use to fully leverage the LLM's capabilities within the $\texttt{GATE}$ framework. This includes a discussion about (1) the metadata-informed prompts, (2) in-context samples provided, as well as (3) methods for aggregating the potential outcomes sampled from the LLM. We have also reeemphasised that the details and templates of the prompts used can be found in the appendix, to ensure the reproducibiity of our results.
>
> **Empirical validation.** We investigate the effect of these prompting strategies on the downstream performance of the CATE models in `Section 6.4`. The results indicate that informing the LLM about the context of the task (`Figure 2`), as well as providing it with in-context samples from $\mathcal{D}^{(obs)}$ (`Figure 4`) helps to significantly improve the downstream performance.
>
> **Action Taken:** We have revised `Section 4.1` to detail how to utilise the LLM within the $\texttt{GATE}$ framework.
>
> ---
>
> ## Connection between theoretical analysis and design choices for the $\texttt{GATE}$ framework
>
> We would like to highlight the following connections between the theoretical analysis (`Section 3`) and the design choices made for the $\texttt{GATE}$ framework (`Section 4`):
>
> 1. **Theoretical Insight:** Tuning the distribution $Q$ via the admissible set $\mathcal{X}_t$ allows to navigate the trade-off between the bias introduced by the generator, and the reduction of variance and covariate shift achieved via data augmentation. In other words, performing data augmentation only in a selected subset of the covariate space $\mathcal{X}_t \subset \mathcal{X}$ can allow to obtain performance benefits even when the generative model is imperfect.
>
> **Design Choice:** We generate the missing potential outcomes only in selected a subset of the covariate space. The exact proportion of potential outcomes generated is modulated by the parameter $\alpha$, and we instantiate $\texttt{GATE}$ with $\alpha = 0.5$ for our experiments (unless otherwise stated).
>
> **Empirical Verification Through Experiment 6.5.2 (Figure 5, middle):** We verify that modulating $\mathcal{X}_t$ allows to navigate this trade-off.

---

> ### Author Response · Authors · 2024-11-22
> **Response to Reviewer iCno (Part 2/3)**
>
> 2. **Theoretical Insight:** Excluding from the admissible set regions of the covariate space where the generative model is particularly "incorrect" (the IPM term measuring the distance between the true distribution $Y(t) | X$ and the generated distribution $Y^{(gen)}(t) | X$ is large) can improve performance.
>
> **Design Choice:** As we explain in `Section 4.2`, assessing the statistical distance between these two distributions is challenging, particularly in the small sample regime and for models trained on external data sources. Thus, we propose to identify such "incorrect" regions using a _proxy measure_: the uncertainty in the generated outcomes. Indeed, in the context of LLMs, it has been shown that uncertainty measures such as variance can be used to discriminate between factually correct and incorrect responses (Huang et al., 2023; Manakul et al., 2023), as well as predict the quality of a response (Lin et al., 2023).
>
> **Empirical Verification Through Experiment 6.5.2 (Figure 5, right):** We verify that as we increase the allowed level of uncertainty of $P_{t, x}^{(gen)}$, the bias introduced by data augmentation increases, while the covariate shift decreases.
>
> ---
>
> ## Datasets used in the experiments
>
> **How are the datasets we use in our experiments created?** Let us clarify that in our experiments (`Section 6`), **none** of our benchmarks dataset is fully synthetic. Indeed, they are obtained as follows:
> - **Lalonde CPS1:** Constructed using a generative model trained on the  **real-world** Lalonde CPS1 dataset. The covariates are the same as in the original dataset, while the treatments and outcomes are obtained with the generative model. This generative model is extensively tested in the RealCause benchmark (Neal et al. 2020) to ensure that it is representative of real-world data.
> - **STAR Project and Hillstrom:** Constructed by subsampling **real-world** randomised controlled trials, following the method described in Gentzel et al. (2021). We obtain the counterfactual outcomes by fitting a 'ground truth CATE model' on the full (significantly larger), randomised version of the datasets, following standard practices (Künzel et al. 2018).
>
> We included more detailed description for how these datasets are constructed in `Appendix D.2` in our manuscript.
>
> **Why are these techniques necessary?** The fundamental challenge of evaluating CATE models lies in the fact that in real-world datasets, the ground truth value of the treatment effect is never observed. This makes the objective evaluation of CATE model performance on a held-out test set impossible when using real-world observational data. To counteract this problem, many different evaluation methods for constructing synthetic or semi-synthetic datasets have been proposed, however all of them have their respective flaws (Curth et al., 2021b).
>
> **What alternative datasets are used in the literature?**
> CATE models are typically evaluated using fully synthetic datasets, where covariates, treatments, and outcomes are all simulated (e.g., Curth & van der Schaar, 2021a; Wager & Athey, 2018), or semi-synthetic datasets like IHDP (Hill et al., 2016), which use real-world covariates but synthetically generated outcomes, using pre-defined parametric functions. Many causal inference papers rely exclusively on such datasets (e.g., Shi et al., 2018; Wager & Athey, 2018; Hassenpour & Greiner, 2018; Jesson et al., 2021), even though they may fail to capture the complexity of real-world problems.
>
> However, semi-synthetic datasets are particularly unsuitable for evaluating the $\texttt{GATE}$ framework, as their outcomes lack grounding in reality, preventing a proper assessment of the LLM's ability to incorporate external knowledge. For this reason, we focused on more reality-grounded datasets and evaluation practices. Nevertheless, for completeness, we also tested LLM-instantiated $\texttt{GATE}$ on the IHDP dataset in `Appendix F.2` to evaluate its few-shot learning capabilities.

---

> ### Author Response · Authors · 2024-11-22
> **Response to Reviewer iCno (Part 3/3)**
>
> ## Significance of improvement when using data augmentation
> We verify that LLM-instantiated $\texttt{GATE}$ offers significant performance improvements compared to the no-augmentation baseline by conducting  two-sample t-tests on the $\sqrt{\epsilon_{PEHE}}$ obtained with and without data augmentation (for the complete datasets, i.e $\rho=1.0$). We report the p-values in the table below, which we also included in `Appendix G.1` of the revised manuscript.
>
>
> | Learner | Lalonde CPS1D | STAR | Hillstrom |
> | -------- | -------- | -------- | -------- |
> | S-learner   | $3.0 \times 10^{-2}$ ★| $2.2 \times 10^{-2}$ ★|$1.3 \times 10^{-2}$ ★|
> | T-learner   | $4.0 \times 10^{-5}$ ★| $4.0 \times 10^{-3}$ ★|$1.3 \times 10^{-5}$ ★|
> | X-learner   | $1.3 \times 10^{-3}$ ★| $1.5 \times 10^{-4}$ ★|$4.1 \times 10^{-3}$ ★|
> | R-learner   | $1.1 \times 10^{-2}$ ★| $1.5 \times 10^{-7}$ ★|$4.9 \times 10^{-4}$ ★|
> | IPW-learner | $1.0 \times 10^{-4}$ ★| $1.9 \times 10^{-5}$ ★|$7.4 \times 10^{-9}$ ★|
> | DR-learner  | $6.8 \times 10^{-6}$ ★| $1.3 \times 10^{-1}$ |$1.4 \times 10^{-4}$ ★|
> | CFR-Wass    | $3.7 \times 10^{-1}$| $3.4 \times 10^{-6}$ ★|$3.8 \times 10^{-1}$|
> | CFR-MMD     | $4.9 \times 10^{-6}$ ★| $1.6 \times 10^{-1}$ |$6.0 \times 10^{-1}$|
> | TARNet      | $1.1 \times 10^{-5}$ ★| $2.8 \times 10^{-1}$ |$3.2 \times 10^{-16}$ ★|
> | DragonNet   | $1.7 \times 10^{-1}$| $7.3 \times 10^{-6}$ ★|$1.8 \times 10^{-7}$ ★|
> | BART        | $1.3 \times 10^{-6}$ ★| $6.6 \times 10^{-2}$ |$3.0 \times 10^{-1}$|
>
> These results demonstrate that performance gains obtained with $\texttt{GATE}$ are statistically significant at the $0.05$ level across the majority of the CATE learners and datasets.
>
> **Action Taken:** To further emphasize the significance of our results, we have included the above table in `Appendix G.1` of the revised manuscript.
>
> ---
>
> ## Other questions
>
> **Training of generative models.** Regarding the baseline generative models trained on $\mathcal{D}^{(obs)}$, we have included the details of how these were trained and set up in `Appendix D.4`. In particular, for the mean imputation model, we set $Y^{(gen)}(t) = \frac{1}{n_t}\sum_{i=1}^n Y_i \mathbb{1} (T_i = t)$, where $(X_i, T_i, Y_i) \in D^{(obs)}_t$ (we compute this mean over the training set only).
>
> **Scoring function.** In the instantiation of $\texttt{GATE}$ we use a percentile-based definition of the scoring function $s(x, t)$ (where choosing $\alpha = 0.5$ guarantees that 50\% of missing potential outcomes are generated, thus allowing to fix the proportion of generated outcomes across datasets. The $\texttt{GATE}$ framework also allows the practitioner to use fixed variance thresholds instead, e.g. defining ($\alpha_{0}$, $\alpha_{1}$) for the control and treated group respectively, and setting $\mathcal{X}_{t} = \\{X_i \mid i \in [n], s(X_i, T_i) < \alpha_t\\}$.
>
> The values of $\alpha_{0}$ and $\alpha_{1}$ could be guided by domain-knowledge or exploratory analysis of the data. Using a fixed threshold would more explicitly guardrail against the inclusion in the augmented dataset of particularly 'poor' generated outcomes, as the proportion of generated outcomes would depend on the properties of the generative model. In particular, if the model is particularly bad, no potential outcomes are generated and our method recovers the baseline performance.
>
> **Actions taken:** We have included a more detailed discussion about the threshold for the scoring function in `Appendix B.3`.
>
>
> ---
>
> *We hope the reviewer’s concerns are addressed and they will consider updating their score. We welcome further discussions.*

---

> ### Author Response · Authors · 2024-11-22
> **Response to Reviewer iCno (References)**
>
> ### References
> Athey, Susan, and Guido Imbens. "Recursive partitioning for heterogeneous causal effects." Proceedings of the National Academy of Sciences 113.27 (2016): 7353-7360.
>
> Curth, Alicia, and Mihaela Van der Schaar. "Nonparametric estimation of heterogeneous treatment effects: From theory to learning algorithms." International Conference on Artificial Intelligence and Statistics. PMLR, 2021a.
>
> Curth, Alicia, et al. "Really doing great at estimating CATE? a critical look at ML benchmarking practices in treatment effect estimation." Thirty-fifth conference on neural information processing systems datasets and benchmarks track (round 2). 2021b.
>
> Dorie, Vincent, et al. "Automated versus do-it-yourself methods for causal inference: Lessons learned from a data analysis competition." (2019): 43-68.
>
> Gentzel, Amanda M., Purva Pruthi, and David Jensen. "How and why to use experimental data to evaluate methods for observational causal inference." International Conference on Machine Learning. PMLR, 2021.
>
> Hassanpour, Negar, and Russell Greiner. "CounterFactual Regression with Importance Sampling Weights." IJCAI. 2019.
>
> Hill, Jennifer L. "Bayesian nonparametric modeling for causal inference." Journal of Computational and Graphical Statistics 20.1 (2011): 217-240.
>
> Jesson, Andrew, et al. "Causal-bald: Deep bayesian active learning of outcomes to infer treatment-effects from observational data." Advances in Neural Information Processing Systems 34 (2021): 30465-30478.
>
> Johansson, Fredrik, Uri Shalit, and David Sontag. "Learning representations for counterfactual inference." International conference on machine learning. PMLR, 2016.
>
> Neal, Brady, Chin-Wei Huang, and Sunand Raghupathi. "Realcause: Realistic causal inference benchmarking." arXiv preprint arXiv:2011.15007 (2020).
>
> Shalit, Uri, Fredrik D. Johansson, and David Sontag. "Estimating individual treatment effect: generalization bounds and algorithms." International conference on machine learning. PMLR, 2017.
>
> Shi, Claudia, David Blei, and Victor Veitch. "Adapting neural networks for the estimation of treatment effects." Advances in neural information processing systems 32 (2019).
>
> Wager, Stefan, and Susan Athey. "Estimation and inference of heterogeneous treatment effects using random forests." Journal of the American Statistical Association 113.523 (2018): 1228-1242.

---

### Author Response · Authors · 2024-11-22
**Global Response**

Based on the reviewers' comments, we have revised our manuscript to clarify and emphasize more prominently the key contributions of our paper:

(1) **Challenging CATE settings:** We focus on settings that are particularly challenging for CATE estimation, namely the small-sample regime and high covariate shift scenarios, where traditional methods often struggle.

(2) **Generalised framework:** We propose a unified framework that generalises existing approaches into a flexible data augmentation solution, allowing to seamlessly integrate any generative model.

(3) **Leveraging external knowledge:** Crucially, our framework enables the use of generative models trained on external data sources, such as LLMs, to address the limitations of observational datasets. This ability is especially significant in the identified challenging settings, as external knowledge provides robust prior information that can compensate for limited data or mitigate the effects of covariate shifts.

(4) **Empirical evidence:** Incorporating insights from external sources leads to consistent performance improvements across all the considered CATE models, and consistently outperforms augmention with generative models trained solely on the observational dataset, particularly in the small-sample regime.

To enhance the presentation of these contributions, we have made the following changes to our manuscript (the revised version of which we have uploaded on OpenReview, highlighting the changes made in teal):

1. **Abstract and Introduction:**  Highlighted the problem that  $\texttt{GATE}$ tackles (CATE estimation in small-sample regimes and covariate shift), and how the general $\texttt{GATE}$ augmentation framework can address it with generative models trained on external sources of information. In particular, we clarified our focus on LLM-based augmentation in this work.
2. **Section 4:** Discussed the key advantages of instantiating  $\texttt{GATE}$ with LLMs. We detailed the design choices associated with this instantiation, including prompting strategies as well as handling the stochasticity of the LLM (`Section 4.1`). In `Section 4.2`, we motivate the variance-based selection heuristic and connect it to the LLM uncertainty quantification litterature.
3. **Section 6:** Reordered the experiments to prioritize the LLM-based instantiation, with a new sensitivity analysis experiment (`Section 6.4`) to analyze the importance of in-context learning abilities of the LLM on downstream CATE performance, complementing the experiment on the importance of context to elicit prior knowledge.


We sincerely thank the reviewers' for their comments which have motivated these changes, which we believe significantly clarify the novelty and contribution of $\texttt{GATE}$, and we remain open to further comments.

With thanks,

The Authors of \#7498

---

### Meta-Review · Area_Chair_fJPT · 2024-12-19

**Metareview:**

The authors develop a novel data augmentation approach, named GATE, to address covariate shifts in Conditional Average Treatment Effect (CATE) estimation. GATE essentially enables the integration of external knowledge into downstream CATE models, by leveraging generative models trained on external data sources, such as large language models (LLMs). The empirical studies evaluate the effectiveness of augmenting a subset of the covariate space, demonstrating that their method generally improves performance. This enhancement is attributed to the flexibility inherent in the GATE framework.

The reviewers have raised concerns regarding substantial overlap with previous work and lack of thorough comparison with other LLMs and generative models. This is a valid point given that this paper is positioned to leverage external LLMs for data augmentation. The rebuttal has addressed some concerns but some reviewers express concern that there is a substantial idea overlap between this work and previous work, which cast doubts on its contribution.

Taking into account all the above points, I think this paper still needs to strengthen its empirical studies to meet the bar for acceptance.

**Additional Comments On Reviewer Discussion:**

Although the authors have attempted to address the reviewers' concerns, the rebuttal has not fully addressed the concerns from the reviewers. Most reviewers have responded to the authors' rebuttal. The post-rebuttal consensus remains leaning on rejection.

There is also the concern that this work has very similar motivation and high-level ideas to previous work. It was raised by one reviewer and I have followed up with the reviewer regarding this (see the discussion thread).

It appears that the authors have updated the manuscript to position reasonably against previous work. Although this paper was originally flagged for ethic review, the reviewer now agrees that this is no longer a severe issue.

---

### Decision · Program_Chairs · 2025-01-22

Reject